# Verbalized Confidence Triggers Self-Verification: Emergent Behavior Without Explicit Reasoning Supervision

## Abstract

Large language models (LLMs) are increasingly used as reasoning partners in domains such as mathematics, coding, and decision support, where reliable expression of confidence is essential for human–AI interaction. However, current LLMs often generate incorrect answers with high confidence, causing significant risks in downstream decision-making. Prior approaches for inducing verbalized confidence, including reinforcement learning or external probing, have shown limited generalization across complex reasoning and unseen tasks. We introduce Confidence-Supervised Fine-Tuning (CSFT), a simple method that trains models to output both an answer and an explicit confidence statement. CSFT substantially reduces calibration errors while also improving accuracy, induces emergent self-verification behaviors such as self-checking under low confidence, and reshapes token distributions into a locally smooth structure around correct answers. Furthermore, although trained only on reasoning tasks, CSFT generalizes to non-reasoning benchmarks such as MMLU. These results establish verbalized confidence as a scalable mechanism for improving calibration, reasoning, and generalization in LLMs.

## 1 Introduction

Large language models (LLMs) have now advanced beyond simple natural language generation to demonstrate expert-level performance in domains such as mathematics and coding, increasingly serving as partners that reason, communicate, and support decision-making with users (Achiam et al., 2023; Zhou et al., 2023; Shao et al., 2024). In such interactive contexts, the importance of verbalized confidence becomes particularly pronounced: model outputs must not only provide correct answers but also explicitly express their own confidence so that users can appropriately interpret and act upon them. However, existing LLMs often generate incorrect outputs with high confidence, and such overconfidence can lead to serious decision-making errors if left undetected by users (Jang et al., 2024). Consequently, enabling models to convey their confidence in a human-interpretable linguistic form has emerged as a critical research direction.

In order to make LLM confidence more interpretable to humans, recent studies have taken two main directions. One line of work leverages reinforcement learning to directly train models to articulate their confidence in words (Band et al., 2024; Stengel-Eskin et al., 2024). Another line instead relies on probing techniques, applying external classifiers to the model's internal states to indirectly recover measures of confidence (Jang et al., 2024; Zhang et al., 2025). However, these methods have not provided convincing evidence of generalization to complex reasoning problems or to unseen tasks. Despite these limitations, verbalized confidence remains an important capability for assessing model reliability and supporting human–AI interaction. Indeed, a line of work (Yoon et al., 2025) reports that reasoning models, often referred to as *System-2* models such as DeepSeek-R1 (Guo et al., 2025) and OpenAI's o1/o3 family, exhibit stronger confidence verbalization capabilities than non-reasoning models. Nevertheless, these studies remain limited to narrow inference scenarios and lack systematic quantitative evaluation across diverse reasoning benchmarks, detailed causal analysis, or an examination of whether post-pretraining methods can lead to further gains.

To address these limitations, we propose a simple but effective fine-tuning approach, Confidence-Supervised Fine-Tuning (CSFT). CSFT trains a model to generate an answer to a given question followed by a natural language statement of its confidence, such as "I am 80% confident." Although our initial goal was to improve calibration, we find that CSFT not only reduces calibration error but also yields substantial gains in accuracy, induces emergent reasoning behaviors, alters the structure of output token distributions, and exhibits cross-setting transferability. Specifically, CSFT simultaneously improves accuracy and reduces Expected Calibration Error (ECE), which measures the calibration error, across a range of reasoning benchmarks. Our analysis reveals two key mechanisms. First, CSFT induces *emergent self-verification* without explicit reasoning supervision: when verbalizing low confidence, the model spontaneously generates self-checking phrases such as "let me recalculate" or "let me double-check" and produces longer outputs, leading to more deliberate reasoning and higher accuracy. Second, CSFT reshapes the token-level output distribution, causing the numeric tokens around the correct answer to form a *locally smooth alignment*. Moreover, CSFT generalizes beyond reasoning tasks: although trained only on reasoning benchmarks, it improves both accuracy and calibration on non-reasoning tasks such as MMLU (Hendrycks et al., 2020). This demonstrates that verbalized confidence is not a reasoning-specific artifact but rather a *transferable representational structure* across language generation.

In summary, CSFT shows that simple confidence supervision alone can yield (i) well-calibrated confidence expression, (ii) self-verifying and more deliberate chain-of-thought reasoning, (iii) locally smoothed output distributions, and (iv) cross-setting transferability, thus serving as a *unified mechanism* for improving both calibration and reasoning in LLMs.

**Contributions.** Our main contributions are as follows:

- We introduce CSFT, a straightforward and effective framework that substantially improves calibration through verbalized confidence supervision.
- We demonstrate that CSFT induces *emergent self-verification*, modulating response length and inserting self-checking phrases by confidence, thereby improving reasoning accuracy.
- We show that CSFT reshapes the output distribution into a *locally smooth structure* around the correct token, providing a mechanistic explanation for the reduction in ECE.
- We establish that CSFT, though trained on reasoning tasks, generalizes to *non-reasoning benchmarks*, exhibiting cross-setting transferability with practical utility.
- Collectively, CSFT provides a unified and scalable mechanism that simultaneously advances calibration, reasoning, distributional structure, and generalization in LLMs.

## 2 RELATED WORKS

### 2.1 CONFIDENCE CALIBRATION IN LLMS

The calibration of LLMs has been examined from multiple perspectives, broadly categorized into likelihood-based methods and approaches centered on verbalized confidence. The former leverage internal model signals, such as token-level entropy or sequence probabilities, to estimate predictive model uncertainty (Desai & Durrett, 2020; Nguyen et al., 2024; Kadavath et al., 2022). While these measures provide useful insights for model-side diagnostics, they do not yield explicit confidence statements that are readily interpretable by humans. This limitation substantially constrains their applicability in high-stakes, user-facing settings, where transparent communication of uncertainty is essential for fostering trust, ensuring accountability, and enabling effective human–AI collaboration.

Verbalized confidence, where models explicitly state their certainty, offers a more user-friendly and interpretable alternative (Lin et al., 2022; Band et al., 2024; Stengel-Eskin et al., 2024). Early attempts like Lin et al. (2022), which predate the adoption of chain-of-thought (CoT) reasoning, applied supervised fine-tuning for arithmetic using fixed labels derived from subtask averages. Critically, this approach forces the model to memorize coarse task difficulty rather than assessing its own instance-level uncertainty. On the other hand, while numerous methods have recently been proposed, approaches are often complex, frequently relying on reinforcement learning or external classifiers, and their application has been largely confined to short-form QA tasks, showing poor generalization to more complex and diverse multi-step reasoning scenarios (Kapoor et al., 2024; Jang et al., 2024;

Zhang et al., 2025). While some "System-2" models exhibit better inherent calibration (Yoon et al., 2025), the development of systematic techniques for effectively and reliably inducing this capability within CoT reasoning remains an open and still underexplored research direction.

## 2.2 SELF-VERIFICATION AND CHAIN-OF-THOUGHT OPTIMIZATION

For LLMs to serve as reliable reasoning partners, the ability to self-verify and systematically correct their own line of thought is paramount (Wang et al., 2023). This has motivated research on optimizing CoT outputs for either conciseness or detailed reflection (Nayab et al., 2024; Team et al., 2025; Guo et al., 2025). A persistent limitation of these approaches, however, is the imposition of a static generation style, compelling the model to remain uniformly verbose or brief. Such rigidity inhibits adaptation of reasoning depth to the specific problem's difficulty or its internal states of uncertainty, an essential component of effective problem-solving and deliberation under cognitive constraints.

This one-size-fits-all approach is suboptimal, as an ideal agent should adapt its reasoning, engaging more deliberately when its confidence is low. Although complex reinforcement learning methods have been employed to encourage such adaptive behaviors (Chen et al., 2023; Zhao et al., 2025; Shafayat et al., 2025), they often suffer from training instability and require carefully designed reward engineering, thereby limiting their practical scalability. By contrast, we demonstrate that a single round of straightforward supervised fine-tuning on confidence labels is sufficient to induce these adaptive CoT behaviors. This provides a simpler, more direct, and highly scalable mechanism for developing reasoning models that are not only more thoughtful but also more reliable in practice.

## 3 CONFIDENCE-SUPERVISED FINE-TUNING

We introduce Confidence-Supervised Fine-Tuning (CSFT), a simple yet effective method for calibrating verbalized confidence in LLMs under reasoning scenarios, without requiring explicit supervision of the reasoning process. CSFT fine-tunes the model to calibrate its verbalized confidence, while also generating a CoT reasoning trace and final answer, ensuring that the reported confidence more faithfully reflects the model's belief in its own correctness. In this way, users receive not only an answer but also a trustworthy confidence estimate, enabling them to better judge how much to rely on the response.

**Formatting**   First, we guided the target LLM to produce both an answer and a corresponding confidence value in a fixed format. This design allows the model to effectively learn the objective function during training, while also ensuring that it outputs verbalized confidence in the same structured format at inference. Specifically, given a query $q$, the decoder of a pre-trained LLM is prompted to generate a structured response that includes: (i) a CoT reasoning trace $r$ and a final answer $a$, enclosed within `<think>` ... `</think>` and `<answer>` ... `</answer>` tags, respectively; followed by (ii) a suffix confidence prompt (see Appendix C), which elicits a discrete confidence score $c \in \{0, 10, \ldots, 100\}$ expressed in `<confidence>` ... `</confidence>` tags. During training, only the confidence score $c$ is directly supervised, while the reasoning $r$ and final answer $a$ remain unconstrained.

### 3.1 TRAINING PROCEDURE

In this section, we describe the training procedure of CSFT. Our goal is to efficiently endow the model with the ability to produce calibrated verbalized confidence, while preserving as much of the pre-trained LLM's learned features as possible. To achieve this, we fine-tune the pre-trained model $f_{\theta_0}$ using the Low-Rank Adaptation (LoRA; Hu et al., 2021) method. Here, $\theta_0$ denotes the fixed pre-trained parameters, and $\theta$ represents these fixed parameters augmented with additional learnable LoRA weights.

**Self-Confidence Label Generation.**   As outlined in (ii) of the above **Formatting** paragraph, our objective is to ensure that the model generates a discrete, well-calibrated confidence score within the prescribed format. To guide the model toward producing such confidence scores, we directly constructed confidence labels for each query $q$ in the training data. Here, we generate the confidence

labels for the training data using the pre-trained model $f_{\theta_0}$. Because the labels are prepared before training, they can be reused during training, minimizing computational cost.

In order to compute the confidence label, we first sample $K$ full generations $\{(r_0^{(i)}, a_0^{(i)})\}_{i=1}^K \sim f_{\theta_0}(\cdot \mid q)$ before training and estimate the empirical accuracy as

$$\hat{p}(q) = \frac{1}{K} \sum_{i=1}^K \mathbb{1}[a_0^{(i)} = a^\star],$$

where $a^\star$ denotes the gold answer. The reason for generating $K$ answers is to examine the distribution formed when the LLM model is queried $K$ times with the same query, thereby assessing how confident the model is about its answers. Since we use model weights $\theta_0$ to generate responses, the dataset can be constructed in advance, and the same labels can be used consistently throughout training, enabling efficient learning. This setup is feasible because, during fine-tuning, we adopt the LoRA method, which preserves the pretrained features while adding the ability to output well-calibrated confidence scores. Consequently, even after training, the model's confidence remains similar to that of the pretrained model. Further, we provide additional results using adaptive label rather than fixed ones in Appendix B.3.

When constructing the label, we first parse the text between the `<answer>` ... `</answer>` tags in the LLM output to extract each $a_0^{(i)}$, and then checked whether it exactly matched the gold answer $a^\star$. Finally, the self-confidence label was obtained by discretizing the empirical accuracy as

$$c = 10 \cdot \lfloor 10 \cdot \hat{p}(q) \rfloor,$$

which maps $\hat{p}(q) \in [0, 1]$ to a confidence value $c \in \{0, 10, \dots, 100\}$.

**Training Objective.**  Our objective is to ensure that the target LLM produces well-calibrated confidence scores. Therefore, during training, we basically do not apply loss to the reasoning trace $r$ or the final answer $a$; instead, we compute the loss solely on the confidence score. Specifically, let $T_c$ denote the token positions corresponding to the entire confidence span, including the `<confidence>` ... `</confidence>` tags. CSFT minimizes the masked cross-entropy loss over these positions:

$$\mathcal{L}_{\text{CSFT}} = - \sum_{t \in T_c} \log p_\theta(y_t \mid y_{<t}, q),$$

where $p_\theta(y)$ denotes the predicted probability of token $y$ under the LLM $f_\theta$. However, fine-tuning may introduce issues such as forgetting or reduced generalization ability. To mitigate this, we optionally add a KL term over the CoT and answer spans (including their respective tags). Denote the corresponding token positions by $T_{\text{KL}}$. This encourages the model to remain close to the pretrained distribution:

$$\mathcal{L}_{\text{total}} = \mathcal{L}_{\text{CSFT}} + \lambda \sum_{t \in T_{\text{KL}}} \text{KL}(p_\theta(t) \parallel p_{\theta_0}(t)),$$

where $\lambda$ is a weighting hyperparameter, and $p_\theta(t)$ and $p_{\theta_0}(t)$ correspond to $p_\theta(y_t \mid y_{<t}, q)$ and $p_{\theta_0}(\cdot \mid y_{<t}, q)$, respectively. Unless otherwise specified, we set $\lambda = 0$.

## 4 MAIN EXPERIMENTS

**Experiment Setup.**  We conduct experiments on both non-reasoning and reasoning models with different training datasets. For the non-reasoning model, we use LLaMA-3.2-3B-Instruct (Llama Team, 2024), while for the reasoning model, we employ Qwen3-4B (Yang et al., 2025). The training data are constructed by sampling $K = 10$ CoT traces and their corresponding answers for each training query $q$. The non-reasoning model is fine-tuned on GSM8K (Cobbe et al., 2021) and HellaSwag (Zellers et al., 2019), whereas the reasoning model is fine-tuned on MATH (Hendrycks et al., 2021). For evaluation, the non-reasoning model is tested on ARC-Challenge (Clark et al., 2018), HellaSwag for commonsense reasoning, MATH-500 (Lightman et al., 2023) and GSM8K for mathematical reasoning, and MMLU (Hendrycks et al., 2020) for general knowledge reasoning. The reasoning model is evaluated on MATH for mathematical reasoning, GPQA-Diamond (Rein et al., 2024) for high-level factual reasoning, and MMLU-Pro (Wang et al., 2024) for advanced knowledge assessment. For math reasoning tasks specifically, we use LLaMA-3.1-8B (Instruct-tuned)

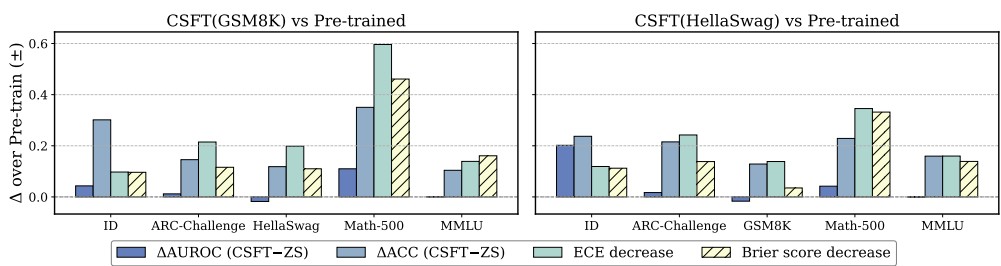

Figure 1: **Comparison of CSFT (trained on GSM8K or HellaSwag) against the pre-trained LLaMA-3.2-3B-Instruct baseline.** Higher bars indicate better performance, where positive values correspond to improvements in ACC and AUROC, or decreases in calibration error (ECE, BS).

only for parsing numeric answers, while all training and evaluation otherwise rely on the designated backbones. See Appendix A for further experimental details.

**Evaluation metrics.** We evaluate all the models in terms of accuracy, area under the reciever-operator curve (AUROC; Hendrycks & Gimpel, 2016), and various calibration metrics such as Expected Calibration Error (ECE; Naeini et al., 2015), Brier Score (BS; Brier, 1950). Throughout this paper, we collectively refer to these metrics as **calibration error**. Details regarding these metrics can be found in Appendix A.1.1.

### 4.1 NON-REASONING MODEL

**CSFT Improves Both Calibration Error and Accuracy.** In Figure 1 (left), LLaMA-3.2-3B with CSFT trained on GSM8K yields consistent improvements across all metrics, not only in calibration error but also in accuracy, even though the model was trained solely on confidence labels (i.e., without providing the correct answer labels or CoT supervision) in the in-distribution (ID) setting. In Figure 1 (right), similar trends are observed when applying CSFT on HellaSwag, where ID accuracy increases by more than 20% while all calibration metrics also improve.

**Generalization Ability.** The calibration and accuracy gains transfer beyond the training domain. For example, as shown in Figure 1, CSFT trained on GSM8K improves calibration error on both ARC-Challenge and HellaSwag, while also increasing accuracy on ARC-Challenge and MATH-500. Similarly, CSFT trained on HellaSwag provides consistent calibration error reductions on ARC-Challenge, GSM8K, and Math-500, and also improves accuracy on ARC-Challenge and MATH-500. These results prove that CSFT generalizes effectively to other domains.

**Transfer to Non-CoT Reasoning Tasks.** CSFT also benefits tasks that do not rely on CoT reasoning, even though it is trained on reasoning cases, as shown in Figure 1. In MMLU, we do not use `<think>` ... `</think>`, but instead directly elicit the answer followed by the confidence query. Nevertheless, we observe substantial reductions in calibration error, confirming that the learned confidence supervision is not restricted to reasoning-style evaluation.

### 4.2 REASONING MODEL

**CSFT Improves Both Calibration Error and Accuracy.** In Table 1, on the ID reasoning dataset MATH, CSFT yields notable gains for the reasoning model in both accuracy and calibration. In particular, AUROC and accuracy improve over the pre-trained baseline, with accuracy increasing by 2.2%, while ECE is reduced by more than half. These results demonstrate that CSFT effectively enhances both predictive performance and reliability in reasoning settings.

**Generalization Ability.** In Table 1, we also evaluate on the held-out dataset in order to explore the generalization capability for reasoning models as well. On GPQA-Diamond, the CSFT-trained model reduces ECE by 17% and achieves a slight increase in accuracy. This indicates, as in the non-reasoning model, that the benefits of CSFT generalize beyond the training domain.

**Transfer to Non-CoT Reasoning Tasks.** On MMLU-Pro without CoT (i.e., the reasoning model in no-thinking mode), accuracy remains essentially unchanged, but calibration improves consider-

Table 1: **Evaluation of Qwen3-4B after CSFT training on MATH.** The GPQA-Diamond and MMLU-Pro datasets are held-out test sets used to assess robustness and transferability. For MMLU-Pro, we report results in the *non-thinking* mode. Boldface highlights the best-performing results.

| Dataset | Setting | AUROC ↑ | ACC ↑ | ECE ↓ | Brier ↓ | NLL ↓ |
|---|---|---|---|---|---|---|
| ID (MATH) | Pre-trained | 0.7300 | 0.6240 | 0.2663 | 0.2689 | 1.9647 |
| | CSFT | **0.8843** | **0.6460** | **0.1027** | **0.1070** | **0.6918** |
| GPQA-Diamond | Pre-trained | 0.7807 | 0.4495 | 0.4030 | 0.3704 | 1.0550 |
| | CSFT | **0.8301** | **0.4590** | **0.2272** | **0.2290** | **0.6536** |
| MMLU-Pro (non-thinking) | Pre-trained | 0.6038 | 0.4037 | 0.4150 | 0.4051 | 1.1141 |
| | CSFT | **0.6258** | **0.4039** | **0.2672** | **0.2979** | **1.1029** |

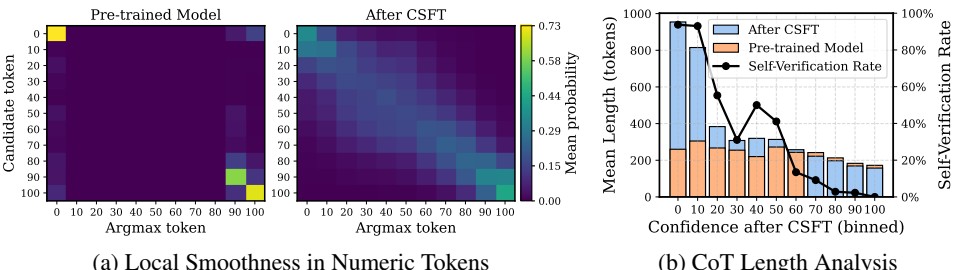

(a) Local Smoothness in Numeric Tokens                (b) CoT Length Analysis

Figure 2: **Emergent behaviors induced by CSFT on LLaMA-3.2-3B trained with GSM8K.** (a) Token-level probability distributions exhibit *local distributional smoothness over numeric tokens*: when predicting a numeric token, nearby tokens receive higher probability mass. (b) Output length and self-verification frequency as a function of confidence: lower-confidence predictions tend to include explicit self-verification behaviors, resulting in longer responses, whereas higher-confidence predictions remain concise.

ably. This suggests that CSFT functions as a general calibration mechanism, transferring its benefits even to non-reasoning tasks, as shown in Table 1.

## 5 ANALYSIS: EMERGENT BEHAVIORS OF CSFT

### 5.1 NUMBER TOKEN SMOOTHING

Despite relying solely on a standard cross-entropy (CE) loss, CSFT induces an emergent local smoothing effect during training, which contributes to improved calibration performance. Conventional CE loss tends to concentrate probability mass exclusively on the target token, failing to capture semantic continuity among numerical tokens that represent confidence levels (e.g., '60', '70', '80'). In LLMs, verbalized confidence is expressed through discrete token sequences, and when the softmax distribution becomes overly sharp, the model struggles to propagate learning signals to semantically adjacent tokens. This leads to poor generalization and weakens the alignment between the generated confidence values and the actual prediction correctness, ultimately degrading calibration errors (Huang et al., 2025).

In contrast, CSFT exhibits a structure-aware smoothing behavior that implicitly mitigates this issue. As shown in Figure 2a, when the model generates a particular confidence number token, it allocates a non-trivial probability mass to nearby numeric tokens, effectively forming a soft neighborhood in the number space. This behavior can be interpreted as the model treating adjacent numbers as partial support for the intended confidence, resulting in a more continuous and interpretable uncertainty representation. This local smoothing effect strengthens the alignment between verbalized confidence and actual accuracy, and enhances the model's ability to express uncertainty in a calibrated manner, even without explicitly modifying the loss function or model architecture. CSFT thus provides a scalable path toward improving LLM calibration by leveraging emergent properties of number token generation.

---

**Low-confidence ⇒ Extended response (Self-verification triggered)**

$q$: The vending machines sell chips for 40 cents and candy bars for 75 cents. George spent $5 and got 3 bags of chips and had 1% of his money left. How many candy bars did he buy?
$a^\star = 5$

**Pre-trained**

`<think>` To solve this problem, we need to first find out how much money George spent on chips. We know that he spent $5 in total and had 1% of his money left. This means he spent 99% of his money, which is $4.90. Since each bag of chips costs 40 cents, we can divide $4.90 by 0.40 to find out how many bags of chips he bought. `</think>`

`<answer>` George spent $4.90 on chips, and each bag costs 40 cents. To find out how many bags he bought, we divide $4.90 by 0.40. $4.90 / 0.40 = 12.25. Since we can't buy a fraction of a bag, we round down to 12 bags of chips. `</answer>`

*Total token #: 95*

**How confident are you in your previous answer?**
`<confidence>` 90 `</confidence>`

**CSFT**

`<think>` To solve this problem, we need to break it down into steps.
... (omitted) ...
To find the number of candy bars George bought, we can divide the amount of money he spent on candy bars (5 cents) by the cost of each candy bar (75 cents): 5 / 75 = 0.0667.
... (omitted) ...
**However, this is not the correct answer.** We should instead find the amount of money George had after buying chips and before buying candy bars.
... (omitted) ...
Now, we can divide this amount by the cost of each candy bar (75 cents) to find the number of candy bars George bought: 375 / 75 = 5.
`</think>`
`<answer>` 5 `</answer>`

*Total token #: 481*

**How confident are you in your previous answer?**
`<confidence>` 10 `</confidence>`

---

**High-confidence ⇒ Shortened response**

$q$: Jack had $100. Sophia gave him 1/5 of her $100. How many dollars does Jack have now?
$a^\star = 120$

**Pre-trained**

`<think>` First, we need to find out how much money Sophia gave to Jack.
Since Sophia also had $100 and gave 1/5 of it to Jack, we can calculate the amount she gave by multiplying 100 by 1/5. Now, we multiply 100 by 1/5, which is the same as dividing 100 by 5. 100 / 5 = 20. So, Sophia gave Jack $20. Next, we need to add the amount Sophia gave to Jack to his initial amount of $100. We can do this by adding 20 to 100. 20 + 100 = 120. Therefore, Jack now has $120. `</think>`
`<answer>` Jack now has $120. `</answer>`

*Total token #: 145*

**How confident are you in your previous answer?**
`<confidence>` 100 `</confidence>`

**CSFT**

`<think>` To find out how many dollars Jack has now, we need to calculate 1/5 of Sophia's $100 and add it to Jack's initial $100. First, we need to find 1/5 of $100. 1/5 of $100 is $100 / 5 = $20. Now, we add $20 to Jack's initial $100. $100 + $20 = $120. `</think>`
`<answer>` $120 `</answer>`

*Total token #: 92*

**How confident are you in your previous answer?**
`<confidence>` 100 `</confidence>`

---

Figure 3: **Confidence reshapes reasoning behavior.** Left: In the low-confidence case (top), the CSFT model generates a long reasoning trace with explicit self-verification, eventually arriving at the correct answer, while the pre-trained model fails. Right: In the high-confidence case (bottom), both models give the correct answer, but the CSFT response is significantly more concise, reflecting confidence-aware brevity.

## 5.2 COT LENGTH OPTIMIZATION

In this section, we analyze how training with CSFT induces self-verification behavior and how this behavior correlates with predicted confidence levels. We further support our analysis with concrete examples. First, Figure 2b presents two key relationships: (1) the average CoT token length as a function of predicted confidence, and (2) the proportion of answers that trigger self-verification across different confidence levels.

The results in Figure 2b show a clear trend: CSFT-trained models generate significantly longer outputs when their confidence is low. In particular, for the lowest confidence bin (0), the average output length is nearly five times longer than that of the pre-trained baseline. This indicates that the model

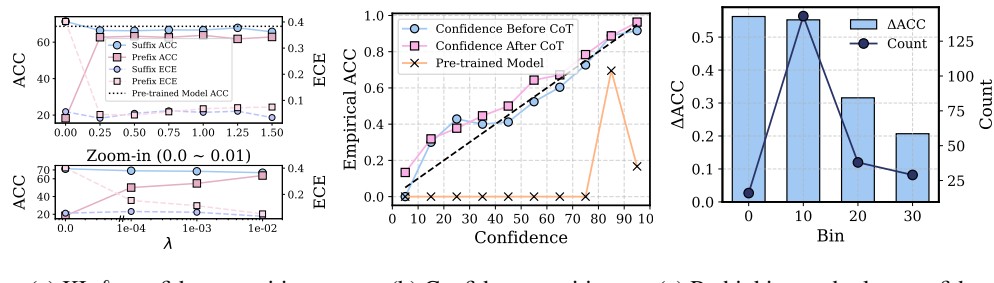

(a) KL & confidence position      (b) Confidence position      (c) Rethinking under low confidence

Figure 4: **Ablation analysis of CSFT using LLaMA-3.2-3B trained on GSM8K.** (a) Effect of KL divergence and confidence prompt position during training. (b) Effect of placing the confidence query before CoT and answer generation. (c) Effect of manually triggering alternative reasoning paths under low-confidence predictions.

compensates for low confidence by engaging in extended reasoning, suggesting that self-verification is an emergent behavior tied to uncertainty. This increase in length is closely accompanied by a high self-verification rate, with nearly all corrected answers involving explicit verification behaviors. The proportion of self-verification was measured by examining CoT traces generated by GPT-4.1-mini, which was prompted to detect the presence of explicit verification behaviors (see Figure 17 for the prompt). Moreover, this self-verification behavior under low-confidence scenarios can serve as a valuable mechanism for end users relying on LLM responses. It provides an implicit signal that the model is uncertain and is actively working to validate its answer, thereby enhancing the perceived reliability of the predicted confidence from the user's perspective. As shown in Appendix B.6, both the length and self-verification patterns generalize beyond the training distribution, emerging similarly on unseen CoT tasks as well.

**Qualitative Examples.** In Figure 3, we provide qualitative examples comparing responses generated by the pre-trained baseline and those from the CSFT fine-tuned LLaMA-3.2-3B on GSM8K. The results highlight that CSFT-trained models dynamically adjust the length and structure of their reasoning based on their predicted confidence. Low-confidence cases elicit longer, reflective traces with internal correction, whereas high-confidence responses tend to be brief and decisive. Additional examples are provided in the Appendix B.

## 6 ABLATION STUDIES

### 6.1 CONDITIONS FOR EFFECTIVE CSFT TRAINING

**Label Quality and Confidence Question.** To evaluate the role of confidence supervision and the design of the confidence prompt in enabling stable training and achieving strong performance, we conduct two ablation studies, summarized in Table 7 of Appendix B. First, we randomly assign confidence labels within the `<confidence>` tag, breaking the link between prediction quality and label supervision. Second, we remove the explicit confidence prompt, instead asking the model to generate a scalar confidence directly after the final answer using a repeated `<answer>` tag. In the first setting, performance significantly degrades across all metrics, confirming that accurate supervision is critical for learning calibrated confidence. In the second setting, the model fails to train altogether, suggesting that without an explicit instruction to predict confidence, the model cannot ground the meaning of the scalar and collapses.

**Prompt Position.** The *suffix* setting, in which the confidence prompt is appended after the model's answer, is described in § 3. In addition, we evaluate a *prefix* variant where the confidence prompt is inserted immediately after the question and before any reasoning begins (see Appendix C). This placement allows the model to condition its reasoning on anticipated confidence, which may affect both generation and calibration. As shown in Figure 4a, the prefix setting consistently yields lower accuracy than the pre-trained baseline, with performance sharply degrading when KL regularization is removed (i.e., $\lambda = 0$). This suggests that without constraints on the CoT and answer spans, the model may overfit to expressing uncertainty rather than reasoning accurately, effectively learning to

be confidently wrong or confidently uncertain. In contrast, the suffix setting shows more favorable behavior. When KL regularization is removed, both calibration and accuracy improve compared to the pre-trained baseline. These results highlight a key trade-off: prefix prompting influences the generation process and requires regularization to remain effective, while suffix prompting is more stable because it does not interfere with the model's reasoning.

**KL Regularization.** Figure 4a We investigate the effect of varying the KL regularization weight $\lambda$ on model performance, focusing on its impact near $\lambda = 0$ (see Figure 4a, Zoom-in figure). The motivation for this analysis is to examine whether performance gains at low $\lambda$ are stable or merely an artifact of tuning. For the prefix setting, we observe that performance rapidly deteriorates as KL regularization is removed. This suggests that, without a constraint to preserve the pretrained distribution over CoT and answer spans, the model may exploit the freedom to optimize ECE at the expense of actual reasoning quality. In effect, the model becomes well-calibrated but confidently incorrect. In contrast, the suffix setting remains stable or even improves in the absence of KL regularization. Since confidence is predicted independently after the full generation, removing the KL constraint does not impair reasoning quality, and may in fact allow for better post hoc alignment of confidence with correctness. These results highlight the importance of controlling model behavior when confidence supervision is introduced at generation time (prefix), as opposed to after-the-fact (suffix).

## 6.2 Inference-Time Ablations

**Effect of Confidence Prompt Position after CSFT.** To test whether the model's confidence relies on observing the length or content of the generated CoT, we compare calibration when confidence is elicited before versus after CoT generation. As shown in Figure 4b, the two reliability curves are broadly similar, suggesting that the model does not depend heavily on CoT visibility and instead bases its confidence on internal uncertainty.

**Confidence-Guided Reasoning Path Refinement.** When a model is well-calibrated, its verbalized confidence serves as a trustworthy signal for downstream decision-making and control. As demonstrated previously, the CSFT-trained model is capable of accurately predicting its confidence even before generating the full CoT. This capability opens up the possibility of guiding the reasoning trajectory from the very beginning.

We exploit this by manually redirecting low-confidence samples toward alternative reasoning paths. Specifically, if the model expresses low confidence in its initial output (elicited via the prefix prompt shown in Figure 11 of Appendix C), we initiate a new reasoning attempt with an altered or more structured prompt (as shown in Figure 16), without waiting for failure. This preemptive rethinking mechanism enables selective refinement with minimal additional cost. As shown in Figure 4c, this confidence-aware rethinking strategy substantially improves accuracy in the low confidence bins. For example, in the 0–10 confidence range, accuracy improves by over 55 percentage points. This result underscores the utility of confidence not just for post hoc calibration, but also for guiding efficient and cost-aware reasoning-time improvement.

## 7 Conclusion

In this work, we introduced Confidence-Supervised Fine-Tuning (CSFT), a simple yet effective method for aligning verbalized confidence in large language models. Our study demonstrates that CSFT not only improves calibration but also enhances accuracy across both reasoning and non-reasoning tasks. Beyond quantitative improvements, CSFT induces emergent behaviors such as self-verification under low confidence and confidence-aware brevity under high confidence, suggesting that verbalized confidence can serve as a reliable control signal for reasoning depth. Furthermore, we show that CSFT reshapes token-level distributions into a locally smooth structure over numeric confidence tokens, providing a mechanistic explanation for improved calibration. Importantly, although trained solely on reasoning datasets, CSFT generalizes to out-of-distribution and non-reasoning benchmarks, highlighting its potential as a unified and transferable mechanism. Taken together, our results establish CSFT as a scalable approach for building more trustworthy language models, bridging the gap between model accuracy, calibration, and user-aligned uncertainty communication.

**Reproducibility Statement.** We present the overall training pipeline in § 3. In addition, Appendix A provides full details of the experimental environment, including model architecture, training hyperparameters, optimization settings, dataset splits, and evaluation metrics, thereby ensuring the reproducibility of our results. We include our code in the supplementary material.

**Ethics Statement.** The proposed method, Confidence-Supervised Fine-Tuning (CSFT), trains models to verbally express confidence alongside their outputs. The datasets used are based on publicly available benchmarks (e.g., GSM8K, MATH-500, MMLU, HellaSwag, GPQA) and do not contain personal or sensitive information. CSFT enables users to explicitly recognize the confidence level of model predictions, which in turn improves the reliability of LLMs in practice and reduces the risk of incorrect decision-making, offering a positive societal impact. *We emphasize that while LLMs were only utilized for minor text polishing, all methods and experiments were fully developed and conducted by the authors.*

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

## A  EXPERIMENTAL DETAILS

### A.1  MODEL AND DATASETS

In this section, we provide detailed information on the models and datasets used in our experiments, along with formal definitions of the calibration metrics employed for evaluation. Specifically, we describe the two instruction-tuned language models used: LLaMA3.2-3B-Instruct, Qwen2.5-1.5B-Instruct, and Qwen3-4B. For datasets, we include:

- **GSM8K** (Cobbe et al., 2021); a dataset of 7.47k grade-school math word problems designed to test step-by-step reasoning. We use 20% of the original training set for validation and the remaining data for training, and the full test set of 1.32k examples for evaluation. Available at https://huggingface.co/datasets/openai/gsm8k.

- **HellaSwag** (Zellers et al., 2019); a commonsense natural language inference dataset consisting of context–completion pairs designed to be challenging for language models. From the original training split (39.9K examples), we randomly sampled 7.5K instances, of which 20% were used as a validation set and the remaining as the training set. We used the official validation split (10K examples) as our evaluation set. Available at https://huggingface.co/datasets/Rowan/hellaswag.

- **MATH** (Hendrycks et al., 2021); a large-scale benchmark of 12.5k competition-level mathematics problems spanning algebra, geometry, number theory, probability, and calculus. We follow the official split, using the training set for model training and the test set for evaluation. Available at https://github.com/hendrycks/math.

- **MATH-500** (Lightman et al., 2023); a subset of the MATH dataset consisting of 500 diverse high school level problems covering algebra, geometry, calculus, and more. Used solely for evaluation. Available at huggingface.co/HuggingFaceH4/MATH-500.

- **ARC-Challenge** (Clark et al., 2018); a multiple-choice science and commonsense QA benchmark containing 1.17k test questions that require reasoning beyond surface-level cues. We use the test set for evaluation. Available at huggingface.co/allenai/ai2_arc.

- **MMLU** (Hendrycks et al., 2020); a 57-task benchmark designed to evaluate knowledge and reasoning across a wide range of academic and professional subjects. We use the official test set containing 14K questions for evaluation. Available at https://github.com/hendrycks/test.

- **MMLU-Pro** (Wang et al., 2024); an improved version of MMLU that emphasizes rigorous evaluation with carefully curated 12K problems to mitigate data contamination and annotation noise. Used solely for evaluation. Available at https://huggingface.co/datasets/TIGER-Lab/MMLU-Pro.

- **GPQA-Diamond** (Rein et al., 2024); a challenging subset of the GPQA benchmark containing 198 graduate-level multiple-choice questions designed to test advanced reasoning and domain expertise. Used solely for evaluation. Available at https://huggingface.co/datasets/Idavidrein/gpqa.

We summarize the hyperparameters used for CSFT fine-tuning in Table 2. Batch size, gradient accumulation steps, and learning rate were tuned over small grids, while other parameters were fixed as listed. We applied KL regularization with $\lambda \in \{0.0, 0.1\}$, and selected checkpoints based on the best development loss. LoRA adapters were injected into the query and value projection matrices with rank 128, scaling factor 32, and dropout 0.1. For all models, we used a maximum input sequence length of 2048 tokens. The only exception was Qwen3-4B, where we set the maximum generation length to 8000 tokens when evaluating on the MATH reasoning benchmark. Although the Qwen3 technical report (Yang et al., 2025) states that the model supports sequences up to 32,768 tokens, GPU resource limitations prevented us from using the full context window. All models were implemented using the Hugging Face transformers library and training was conducted on NVIDIA A6000 GPUs.

Table 2: Training hyperparameters used for CSFT fine-tuning across both models.

| Hyperparameter | Value |
|---|---|
| Batch size | [1, 2] |
| Gradient accumulation | [16, 32] |
| Learning rate | [1e-5, 1e-03] |
| Optimizer | AdamW |
| Weight decay | 0.0 |
| Warmup ratio | 0.0 |
| Max sequence length | 2048 |
| KL regularization ($\lambda$) | [0.0, 0.1] |
| Training steps | 2500 |
| Checkpoint selection | Best dev loss |
| **LoRA configuration** | |
| LoRA rank ($r$) | 128 |
| LoRA alpha | 32 |
| LoRA dropout | 0.1 |
| LoRA target modules | $q_{\text{proj}}, v_{\text{proj}}$ |

### A.1.1 CALIBRATION METRICS

- **Expected Calibration Error** (ECE; Naeini et al., 2015):

$$\text{ECE} = \sum_{m=1}^{M} \frac{|B_m|}{n} \left| \text{acc}(B_m) - \text{conf}(B_m) \right|$$

where $B_m$ is the set of predictions in bin $m$, $\text{acc}(B_m)$ is the accuracy, and $\text{conf}(B_m)$ is the average confidence of the predictions in that bin. ECE measures how well the model's predicted probabilities are calibrated.

- **Brier Score** (BS; Brier, 1950):

$$\text{BS} = \frac{1}{N} \sum_{i=1}^{N} (f_i - y_i)^2$$

where $f_i$ is the predicted probability and $y_i$ is the true label. BS combines both the accuracy and confidence of the predictions, penalizing overconfident and underconfident predictions.

## B ADDITIONAL RESULTS

### B.1 INSTANCE-LEVEL VS. TASK-LEVEL CALIBRATION: CONTRAST WITH LIN ET AL. (2022)

To empirically demonstrate the advantages of our instance-level calibration over prior task-level approaches, we conducted a direct comparison with the method proposed by Lin et al. (2022). While Lin et al. (2022) originally operated in a pre-CoT setting focusing on simple arithmetic, we adapted their methodology to our reasoning benchmark for a fair comparison. Specifically, we trained a LLaMA-3.2-3B-Instruct model on the MATH dataset using their protocol: grouping training examples by their coarse subtask categories (e.g., Algebra, Geometry) and assigning the *average empirical accuracy* of the subtask as the fixed confidence label for all instances within that category.

**Methodological Divergence.** The fundamental difference lies in the granularity of supervision. The approach of Lin et al. (2022) essentially trains the model to memorize *statistical task difficulty* ($P(\text{Correct}|\text{Task})$). As noted in their work, this forces the model to assign low confidence even to trivial instances (e.g., simple calculations) if they fall under a historically difficult category. In contrast, CSFT utilizes $K$-sampling to estimate the *instance-level posterior probability*

($P$(Correct|Instance)), teaching the model to assess its own internal knowledge boundary regardless of the general task difficulty.

**Results and Analysis.** Table 3 presents the performance of both methods across In-Distribution (MATH) and Out-of-Distribution (GSM8K, HellaSwag, MMLU) benchmarks. The results highlight three critical findings:

1. **Failure of Task-Level Generalization:** The method of Lin et al. (2022) collapses on OOD tasks, exhibiting AUROC scores near 50% (e.g., 50.00% on GSM8K, 47.85% on HellaSwag), which is equivalent to random guessing. This confirms that learning "task difficulty" is a dataset-specific bias that does not transfer to new domains.

2. **Robustness of Instance-Level Signals:** CSFT maintains high AUROC and low ECE across all datasets. Notably, on the non-mathematical MMLU benchmark, CSFT achieves an AUROC of 56.70% compared to 40.73% for the baseline, demonstrating that the internalized concept of confidence transfers even to non-reasoning tasks.

3. **Calibration Error:** CSFT significantly reduces the Expected Calibration Error (ECE) and Brier Score compared to the baseline, proving that instance-level supervision is essential for reliable uncertainty estimation in complex reasoning models.

Table 3: **Comparison between CSFT and Lin et al. (2022).** Both models were trained on the MATH dataset. Lin et al. (2022) uses static labels based on subtask difficulty, while CSFT uses dynamic instance-level labels. CSFT demonstrates superior generalization across all OOD benchmarks. (↑: higher is better, ↓: lower is better).

| Dataset | Method | AUROC (↑) | ACC (↑) | ECE (↓) | Brier (↓) |
|---|---|---|---|---|---|
| **MATH** (ID) | Lin et al. (2022) | 53.82 | 28.18 | 0.1813 | 0.2764 |
| | **CSFT (Ours)** | **78.74** | **30.52** | **0.0324** | **0.1638** |
| **GSM8K** (OOD) | Lin et al. (2022) | 50.00 | 67.85 | 0.3215 | 0.3214 |
| | **CSFT (Ours)** | **74.80** | **73.69** | **0.2243** | **0.1805** |
| **HellaSwag** (OOD) | Lin et al. (2022) | 47.85 | 49.36 | 0.2572 | 0.3883 |
| | **CSFT (Ours)** | **79.29** | **69.55** | **0.1787** | **0.2564** |
| **MMLU** (OOD) | Lin et al. (2022) | 40.73 | 48.44 | 0.3043 | 0.4090 |
| | **CSFT (Ours)** | **56.70** | **61.18** | **0.1420** | **0.2787** |

## B.2 VERBALIZED CONFIDENCE VS. DISCRIMINATIVE CLASSIFIERS

While it is true that classifiers can achieve low calibration error on in-distribution (ID) data, the goal of this work is not merely the minimization of a simple metric, but the establishment of *intrinsic verbalized confidence*. Unlike probing methods that treat the model as a black box, CSFT compels the model to internalize the concept of uncertainty, leading to structural changes such as number token smoothing (see Figure 2a). This internal correction is the key driver behind the robust cross-task generalization that external classifiers fail to achieve.

Fundamentally, classifiers operate as external observers that map frozen representations to correctness probabilities without correcting the underlying model's understanding of uncertainty itself. In contrast, CSFT compels the model to explicitly process and generate uncertainty as part of its language modeling objective. This induces an "uncertainty concept correction" within the model. As evidenced by the emergent local smoothing of numeric tokens, the model learns a continuous manifold of confidence. This is not a simple calibration trick, but a fundamental restructuring of the latent space that classifiers cannot replicate.

Consequently, because CSFT modifies the model's internal reasoning-uncertainty alignment, the learned calibration capability becomes a transferable skill. Conversely, classifiers tend to overfit to the specific feature distributions of the training task (e.g., GSM8K). When the domain shifts (e.g., moving to MMLU) and those features change, the classifier's performance collapses. CSFT

generalizes robustly because the "concept" of evaluating one's own confidence is preserved across domains.

As suggested, we implemented both Shared and Separate Classifiers as baselines to validate this. The results in Table 4 confirm that while classifiers can mimic CSFT's performance on the training set (ID), their performance drops significantly on Out-of-Distribution (OOD) tasks compared to CSFT, validating the necessity of our generative approach.

Table 4: **Relative performance gains of CSFT and Classifiers over the pre-trained baseline.** While classifiers show competitive calibration on the ID task (GSM8K), they fail to generalize to OOD tasks, often degrading performance (positive ECE/Brier $\Delta$), whereas CSFT consistently improves metrics. ($\uparrow$: higher is better, $\downarrow$: lower is better).

| Dataset | Method | AUROC ($\Delta \uparrow$) | ACC ($\Delta \uparrow$) | ECE ($\Delta \downarrow$) | Brier ($\Delta \downarrow$) |
|---|---|---|---|---|---|
| **GSM8K** (ID) | **CSFT (Ours)** | **+0.0432** | **+0.3014** | **−0.0973** | **−0.0963** |
| | Shared Classifier | – | – | −0.1149 | −0.0594 |
| | Separate Classifier | – | – | −0.1039 | −0.0252 |
| **Math-500** (OOD) | **CSFT (Ours)** | **+0.1100** | **+0.3503** | **−0.5962** | **−0.4611** |
| | Shared Classifier | – | – | −0.4443 | −0.3578 |
| | Separate Classifier | – | – | −0.1246 | −0.1446 |
| **HellaSwag** (OOD) | **CSFT (Ours)** | −0.0178 | **+0.1184** | **−0.1983** | **−0.1100** |
| | Shared Classifier | – | – | +0.0183 | +0.0615 |
| | Separate Classifier | – | – | +0.0778 | +0.0532 |
| **MMLU** (OOD) | **CSFT (Ours)** | **+0.0006** | **+0.1041** | **−0.1390** | **−0.1608** |
| | Shared Classifier | – | – | −0.0290 | −0.0176 |
| | Separate Classifier | – | – | +0.0101 | +0.0452 |

### B.3 ADAPTIVE LABEL TRAINING

**Adaptive label.** In the adaptive label case, to compute the confidence label, we repeatedly generate $K$ responses $\{(r^{(i)}, a^{(i)})\}_{i=1}^{K} \sim f_\theta(\cdot \mid q)$ during training using the current parameters $\theta$, and construct the label as

$$\hat{p}(q) = \frac{1}{K} \sum_{i=1}^{K} \mathbb{1}\big[a^{(i)} = a^\star\big].$$

As this equation shows, the confidence label reflects in real time how often the current model answers correctly out of $K$ attempts for query $q$. This allows the verbalized confidence to capture the model's actual reliability more faithfully, thereby improving the accuracy of confidence learning. However, this method has the drawback of requiring additional inference runs multiple times during training. In our experiments, we used Fixed label setting as default and conduct an ablation study using Adaptive label setting.

We further compare adaptive versus fixed confidence labeling using CSFT on HellaSwag with the LLaMA-3.2-3B. Table 5 reports the performance differences relative to the fixed-label setting. Overall, the adaptive strategy yields only marginal improvements. These results suggest that adaptive labels, while more faithful to the model's evolving reliability, do not consistently outperform fixed labels. Given the added computational overhead of generating labels during training, the fixed-label approach provides a more efficient trade-off in practice.

Table 5: Comparison of adaptive versus fixed confidence labels for CSFT on HellaSwag with LLaMA-3.2-3B. Values indicate differences relative to the fixed-label setting.

| Dataset | AUROC | ECE | BS |
|---|---|---|---|
| ID | -0.09 | +0.04 | +0.04 |
| MATH | +0.01 | -0.01 | -0.07 |
| MMLU | +0.06 | -0.01 | -0.01 |

Table 6: Calibration results of Qwen2.5-1.5B-Instruct on the in-distribution GSM8K and two held-out reasoning benchmarks. For datasets, ✓: 'seen' during CSFT, ✗: 'unseen' during CSFT. For the metrics, ↓: lower is better and ↑: higher is better.

| Dataset | Method | AUROC (↑) | ACC (↑) | ECE (↓) | BS (↓) |
|---------|--------|-----------|---------|---------|--------|
| **GSM8K (✓)** | Pre-trained | 49.59 | 67.85 | 0.1928 | 0.2915 |
| | CSFT | 67.67 | 69.63 | 0.0552 | 0.2285 |
| **MATH-500 (✗)** | Pre-trained | 59.91 | 55.00 | 0.3786 | 0.2978 |
| | CSFT | 60.27 | 56.40 | 0.2590 | 0.2629 |
| **ARC-Challenge (✗)** | Pre-trained | 54.08 | 52.07 | 0.1660 | 0.2680 |
| | CSFT | 61.63 | 56.82 | 0.1107 | 0.2584 |

### B.4 ADDITIONAL STRUCTURED MODEL

We additionally evaluate the non-reasoning model Qwen2.5-1.5B-Instruct. While detailed results are provided in Table 6, we highlight only the most notable findings here. CSFT yields clear improvements in calibration metrics across datasets, demonstrating that the benefits of CSFT extend to non-reasoning models as well.

### B.5 LABEL QUALITY AND CONFIDENCE PROMPT

Table 7: Ablation analysis on the impact of confidence label quality and the inclusion of confidence prompt. Results are reported as differences relative to CSFT on GSM8K using LLaMA3.2-3B-Instruct.

| Variant | ACC | ECE | Avg. Len. |
|---------|-----|-----|-----------|
| w/o Correct label | -2.14 | +0.05 | -49.36 |
| w/o Conf question | Training collapsed | | |

To assess the importance of confidence supervision and the design of the confidence prompt for stable training and reliable performance, we conduct two ablation studies, summarized in Table 7.

First, we randomize the confidence labels within the `<confidence>` tag, thereby disrupting the alignment between prediction quality and supervision. This results in notable performance degradation across all metrics, highlighting that accurate supervision is essential for learning calibrated confidence.

Second, we remove the explicit confidence prompt and instead require the model to output a scalar confidence immediately after the final answer using a repeated `<answer>` tag. In this case, training collapses entirely, indicating that without explicit prompting, the model cannot properly ground the notion of confidence and fails to learn a meaningful signal.

### B.6 LENGTH ANALYSIS ON HELD-OUT CoT TASKS

Figure 5 and Figure 6 present an analysis of model outputs on Math-500 and ARC-Challenge—two held-out CoT tasks not seen during CSFT training. As shown in both figures, output length increases in low-confidence bins. In the case of Math-500, there is also a clear trend toward more concise responses in high-confidence bins. These results demonstrate that the length modulation effect reported in the main paper is not restricted to the training distribution but generalizes to unseen tasks. In other words, CSFT enables the model to internalize the ability to adjust response length based on its uncertainty, suggesting a deeper transformation in its reasoning behavior.

### B.7 EMERGENT SELF-VERIFICATION IN THE CODING DOMAIN

We additionally evaluated Llama-3.2-3B CSFT, trained on GSM8K, on the MBPP dataset (Austin et al., 2021), a benchmark consisting of 974 crowd-sourced Python programming problems. While

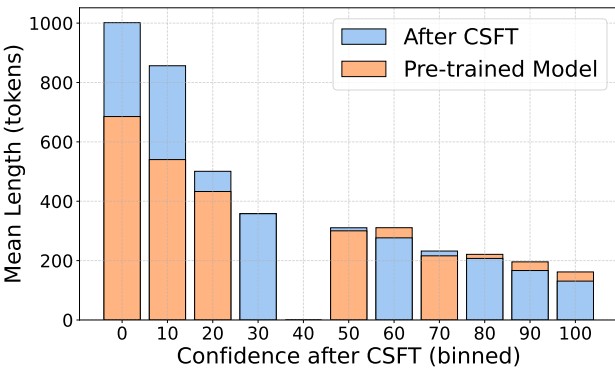

Figure 5: Output length across confidence bins on Math-500, using `LLaMA3.2-3B-Instruct` fine-tuned with CSFT. The model generates significantly longer responses when confidence is low, while high-confidence predictions tend to be more concise.

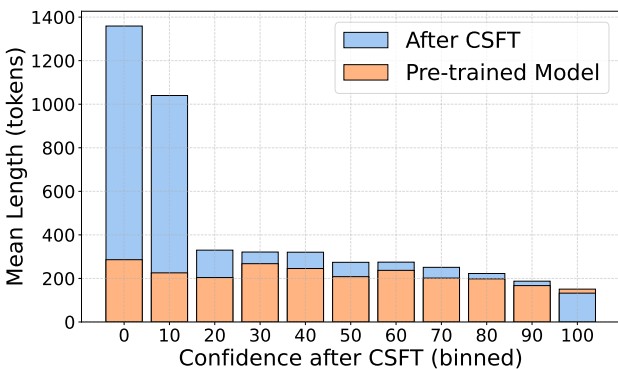

Figure 6: Output length across confidence bins on ARC-Challenge, using LLaMA3.2-3B-Instruct fine-tuned with CSFT. The same trend of confidence-conditioned length modulation emerges, despite the task being unseen during training.

accuracy declined due to format mismatches, we consistently observed self-verification behaviors such as proposing multiple solutions and assessing their correctness or efficiency. As shown in Figures 7 and 8, CSFT exhibits emergent self-verification in code generation: instead of producing a single solution like the pre-trained baseline, the model explores alternatives, critiques them, and refines its outputs.

## C  PROMPT EXAMPLES

This section presents the prompt templates used in our experiments for eliciting model reasoning, answers, and confidence scores. We include both the *prefix* prompt, where the confidence is generated before reasoning begins, and the *suffix* prompt, where confidence is predicted after the final answer. All prompts follow a standardized format to ensure consistent supervision during CSFT and reliable evaluation during inference.

---

**Comparison of `opposite_Signs` Implementations**

**Pre-trained**

```python
def opposite_Signs(a, b):
    return (a >= 0) != (b >= 0)
```

**CSFT**

```python
def opposite_Signs(a, b):
    return (a >= 0) != (b >= 0)
```

Alternatively, you can use the 'math.copysign' function:

```python
import math
def opposite_Signs(a, b):
    return math.copysign(1, a) != math.copysign(1, b)
```

---

Figure 7: **Pretrained vs CSFT on code generation.** The pretrained model provides only a single straightforward implementation, while the CSFT model not only reproduces the same solution but also suggests an alternative formulation using `math.copysign`, illustrating its tendency to generate diverse implementations.

---

**Comparison of `is_octagonal` Implementations**

**Pre-trained**

```python
def is_octagonal(n):
    return (8 * n**2 - 4 * n + 1) // 2
```

**CSFT**

```python
def is_octagonal(n):
    a = 2
    b = 4
    while True:
        a, b = b, 8 * a + 4 * b
        if a >= n:
            return b - 4 * a + 4
        if a == n:
            return b
```

However, the above solution is not efficient. Here's a more efficient solution using the formula for the nth octagonal number:

```python
import math
def is_octagonal(n):
    return (8 * n * (4 * n - 1) - 1) // 6
```

---

Figure 8: **Pretrained vs CSFT on code generation.** The pretrained model proposes an incorrect closed-form expression for the octagonal number. In contrast, the CSFT model explores an iterative update rule and eventually provides a more efficient and mathematically valid closed-form formula.

---

**»> Base Reasoning Prompt**

This is a conversation between **User** and **Assistant**.
The User asks a question, and the Assistant provides a solution.
Before answering, the Assistant reasons through the problem step-by-step.
The reasoning is enclosed within `<think> ... </think>`, and the final answer within
`<answer> ... </answer>`.

**Example:**
`{question}`
`<think>{step-by-step reasoning}</think>`
`<answer>{final answer}</answer>`

Now, respond to the following using the **exact same format**:
`<question>`

Figure 9: Base prompt format used to elicit step-by-step reasoning and final answer generation.

---

**»> Suffix Confidence Prompt**

Please respond with a score from 0 to 100 in `<confidence> </confidence>` tags.
How confident are you in your previous answer?

Figure 10: Suffix confidence prompt used to elicit model confidence *after* the full reasoning and answer generation. Since confidence is predicted post hoc, this setup allows for calibration without interfering with the model's original reasoning or final answer.

---

**»> Prefix Confidence Prompt**

This is a conversation between **User** and **Assistant**.
The User asks a question, and the Assistant provides a thoughtful, reasoned answer.
Before answering, the Assistant first reasons through the problem step-by-step.
The reasoning is enclosed in `<think> ... </think>` tags.
The final answer is enclosed in `<answer> ... </answer>` tags.
A confidence score is then provided in `<confidence> ... </confidence>` tags, representing the Assistant's certainty as a **continuous value between 0 and 100**.

**Example:**
`{question}`
`<think>{step-by-step reasoning}</think>`
`<answer>{final answer}</answer>`
`<confidence>{confidence}</confidence>`

Now, answer the following in **exactly** the same format:
`<question>`

Figure 11: Prefix confidence prompt used to elicit model confidence *before* reasoning begins. By conditioning the generation on anticipated confidence, this prompt not only guides the model's uncertainty expression but also influences the reasoning path and final answer.

**»> GSM8K Parsing Prompt**

**Instruction:**
We have a user's question and a model's generated response:
**Your task:**
1. Carefully read the question and the generated response in **Example 6 only**.
2. Extract the final answer based on the following rules:

- If the response contains a number (with or without units), **extract only the numeric value**.

- If the response is purely textual (no numbers), **extract the exact string as it appears**.

3. Use the following output format: `Model's Final Answer is: [Your extracted answer]`

**Rules:**

- Only process **Example 6** for extraction. Ignore all other examples.

- Do not include units, symbols, or extra text when extracting numbers.

- Provide the answer strictly in the requested format without additional explanations.

**Examples**
**Example 1:** Model's Generated Response: It takes about 160 minutes.
`Model's Final Answer is: 160`
**Example 2:** Model's Generated Response: The nearest star is approximately 4.24 light years away.
`Model's Final Answer is: 4.24`
**Example 3:** Model's Generated Response: The tallest mountain is Mount Everest.
`Model's Final Answer is: Mount Everest`
**Example 4:** Model's Generated Response: It weighs 5 kg.
`Model's Final Answer is: 5`
**Example 5:** Model's Generated Response: 81 + 221 - 24 = 278.
`Model's Final Answer is: 278`

**Example 6:** Model's Generated Response: `<answer_text>`

Figure 12: Prompt used to extract final answers from model-generated responses on GSM8K.

>> **Math500 Matching Prompt**

**Instruction:**
You are given the true answer and the final answer generated by a model for a math problem.
**Your task:**

1. Only examine **Example 6**.

2. Compare the **model's final answer** and the **true answer**.

3. Respond with "yes" if they exactly match, otherwise respond with "no".

4. Do not include any explanation or extra words—just respond with "yes" or "no".

**Examples**
**Example 1:**
True Answer: 0.5
Model Answer: 1/2
Is it correct?: yes
**Example 2:**
True Answer: 24
Model Answer: 22
Is it correct?: no
**Example 3:**
True Answer: 8
Model Answer: 32 / 4 = 8
Is it correct?: yes
**Example 4:**
True Answer: \frac{10}{4}
Model Answer: \frac{9}{4}
Is it correct?: no
**Example 5:**
True Answer: 3
Model Answer: \frac{15}{5}
Is it correct?: yes

**Example 6:**
True Answer: <true_answer>
Model Answer: <model_answer>
Is it correct?:

Figure 13: Matching prompt for evaluating exact answer agreement on Math500. Designed to assess correctness by comparing model output with the ground truth in a strict yet interpretable format.

>> **ARC-Challenge Parsing Prompt**

**Instruction:**
A user's question provides four choices formatted exactly as:

```
A. <option A>
B. <option B>
C. <option C>
D. <option D>
```

We also have the model's generated response.
**Your task:**

1. Read **Example 6 only**.

2. Decide which single choice (`A`, `B`, `C`, or `D`) the model ultimately selected, following these rules:

   - **Letter match** – If the response explicitly includes the letter 'A', 'B', 'C', or 'D' (optionally followed by punctuation), extract **only that letter**.
   - **Text match** – If no letter is given, compare the response text (case-insensitive, ignoring punctuation and surrounding spaces) with each option; if it matches exactly one, return the corresponding letter.
   - If both a letter and option text appear, treat the letter as authoritative.

3. Output format (strict):
   ```
   Model's Final Answer is:  <A | B | C | D>
   ```
**Do not add explanations or any extra text.**

**Examples**
**Example 1:**
Choices: A. Paris    B. Berlin    C. Madrid    D. Rome
Model's Generated Response: A. Paris is the capital of France.
```
Model's Final Answer is:  A
```
**Example 2:**
Choices: A. 3    B. 4    C. 5    D. 6
Model's Generated Response: The correct option is B.
```
Model's Final Answer is:  B
```
**Example 3:**
Choices: A. Spring    B. Summer    C. Autumn    D. Winter
Model's Generated Response: It's usually coldest in winter.
```
Model's Final Answer is:  D
```

**Example 6:**
Choices: `{choices}`
Model's Generated Response: `{answer_text}`

Figure 14: Parsing prompt for multiple-choice answer extraction on ARC-Challenge. The rules prioritize explicit letter selection, with fallback to semantic string matching.

>> **Pre-CoT Confidence Prompt**

Before generating your answer, can you first assess your internal confidence (0–100) in its correctness and state it using '<confidence> </confidence>' tags, then proceed to provide your full answer?

Figure 15: Prompt format for eliciting a model's self-assessed confidence **prior to** generating CoT response.

>> **Low Confidence Rethinking Prompt**

Your confidence score is low. Rather than following your current reasoning path, pause and explore an alternative approach that is likely to raise your confidence. Think step-by-step and provide a revised answer.

Figure 16: Prompt used when the model reports low confidence, encouraging it to pause and reconsider its reasoning path to generate a more confident response.

>> **Self-verification Check Prompt**

You are given a response generated by a language model.

Determine whether the response shows signs of **self-verification**, such as rechecking its answer, correcting previous steps, expressing uncertainty or confidence, or using phrases like "let me recalculate," "this is incorrect," "I double-checked," "I'm not sure," "however," "on second thought," or "this seems off."

Also count it as self-verification if the model initially gives one answer but later changes or revises it based on its own reasoning.

Respond with "yes" if the response includes any indication of self-verification. Otherwise, respond with "no".

Model Response: {model_response}

Answer:

Figure 17: Prompt used to evaluate whether a generated CoT includes explicit self-verification behaviors.

