# OpenReview forum: "Verbalized Confidence Triggers Self-Verification: Emergent Behavior Without Explicit Reasoning Supervision"
_ICLR.cc/2026/Conference — ICLR 2026 Conference Desk Rejected Submission_

### Official Review · Reviewer_V8Nd · 2025-10-28

**Soundness:** 2
**Presentation:** 2
**Contribution:** 1
**Rating:** 2
**Confidence:** 4

**Summary:**

This paper presents confidence-supervised fine-tuning (CSFT), a training paradigm where models are trained to express verbalized confidence. The ground truth confidence for a particular question is approximated as the empirical accuracy of the model on that task. The model is then optimized to output well-calibrated confidence statements. Optimizing with this simple objective improves both accuracy and calibration, and leads to emergent CoT self-checking reasoning.

**Strengths:**

- **Simple, relevant and well-motivated**: The proposed method is simple yet tackles the important problem of model calibration. Calibration is increasingly critical for deploying reasoning models in real-world applications.

- **Interesting results**: Despite its simplicity, the method appears to produce notable and somewhat surprising empirical results (e.g., accuracy improvements), which merit deeper investigation.

**Weaknesses:**

- **Novelty**: It is unclear what the novelty of this framework is. There have been previous works from as long as 3 years ago [1] which use a very similar setup.
- **Experimental Design and Strength**: No baselines (except a simple pre-trained model) are presented in the results, making it difficult to identify the strengths of the method relative to existing works in the field. Some results presented in this paper are also surprising, and merit deeper analysis. In particular, it is unclear to me how this method improves accuracy without optimizing for it in any way.
- **Writing**: The writing of the paper is informal at times and can be significantly improved. For example “we basically do not apply loss to the reasoning trace” (Line 185) makes it unclear whether a loss to the reasoning trace is being applied or not.

[1]: Lin, S., Hilton, J., & Evans, O. (2022). Teaching models to express their uncertainty in words. arXiv preprint arXiv:2205.14334.

**Questions:**

- **Confidence supervision**: Why is empirical accuracy used as a proxy for confidence? This approach marginalizes out the reasoning trace, which is often key to determining confidence (e.g., models may realize mid-reasoning that a solution is correct).

- **Relation to classifiers**: How does this method differ from training a standard classifier that predicts correctness given a reasoning trace? If the model simply outputs a confidence score without reasoning about it, the setup seems equivalent to classification fine-tuning.

- **Novelty**: How does this work differ from prior studies (e.g., [1]) that fine-tuned verbalized confidence using accuracy supervision? How exactly is this different from training a classifier?

- **Accuracy Improvement**: Given that no loss is applied to reasoning traces or answers, why does accuracy improve? A deeper investigation of why this is happening is important, as this is an unintended by-product. Please clear my understanding if I am wrong, but since there is no optimization over the reasoning traces and final answer, and there is a KL to the original model, then optimizing such an objective should not alter the reasoning traces or accuracy at all. Also, the claim of accuracy improving does not seem to be generally substantiated. The accuracy improvements in table 1 for the Qwen model are negligent, suggesting that this trend does not hold generally.

- **Confidence variance**: What is the variance of predicted confidence across different reasoning traces or answers for the same question?

- **Confidence-Guided Reasoning Path Refinement**:  Please provide more details on this. Is this method only applicable for the prefix confidence variant? If a model is prompted for confidence directly after the question, then how will resampling when it outputs low confidence increase accuracy, even if the prompt is different? The resampling prompt asks it to consider an alternative approach but based on my understanding, it has not considered any approach yet (since the confidence was directly after the question). What does alternate refer to in such a scenario? Instead, a better alternative might be to do sequential sampling with the suffix prompt. If the confidence turns out to be low, reprompt the model with its original attempt and predicted confidence, and ask it to generate a new answer with higher confidence.

- **Figure 1:** Fig 1 is not a good way to present results. The bars in the graph represent different metrics and thus the deltas are not easily comparable. Table 1 is a better way to summarize these results.

- **Baselines**: Please add more baselines to allow placing this work relative to existing work:
     - *Shared Classifier*: Train a classifier on top of the reasoning model. This baseline should also have the same accuracy as the pre-trained model, but it uses representations of the model which is generating the answers.
     - *Separate Classifier*: Train a separate classifier model conditioned on generations from the reasoning model. This baseline should have same accuracy as the pre-trained model.

[1]: Lin, S., Hilton, J., & Evans, O. (2022). Teaching models to express their uncertainty in words. arXiv preprint arXiv:2205.14334.

---

> ### Author Response · Authors · 2025-11-19
>
> # [Q1] Confidence Supervision: Why Empirical Accuracy?
>
> We employ empirical accuracy as a theoretically grounded and scalable estimator of epistemic uncertainty. While process-level supervision (e.g., PRMs) faces severe data scarcity and cost bottlenecks, our outcome-based approach aggregates the model’s output distribution to provide a stable supervision signal. Crucially, this method does not marginalize reasoning; rather, it effectively aligns the model's verbalized confidence with the reliability of its reasoning process, inducing emergent self-verification behaviors.
>
> ## (a) Mitigating Stochasticity with Distributional Estimates
>
> Reasoning traces in LLMs are inherently stochastic; a single trace may be correct or incorrect due to random sampling variance, making it a noisy target for confidence supervision. By sampling $K$ traces and computing empirical accuracy, we approximate the true posterior probability of correctness ($P(\text{Correct}|x)$). This effectively marginalizes out the noise of individual sampling paths to capture the model's stable epistemic uncertainty.
>
> ## (b) Capturing Model-Level Uncertainty vs. Trace-Level Heuristics
>
> Confidence is fundamentally a property of the model's knowledge state regarding the input, rather than a feature of a specific text string. While models may sometimes realize a mistake mid-trace, supervising based on a single trace forces the model to overfit to that specific realization. Our distributional approach teaches the model to predict the likelihood of generating a correct answer, which is a robust objective for generalizable calibration.
>
> ## (c) Emergent Reasoning, Not Marginalization
>
> The reviewer expressed concern that this approach might marginalize the reasoning trace. On the contrary, our results demonstrate that supervising with this distributional signal actively reshapes the reasoning process. Because the model learns to align its internal state with its probability of success, we observe emergent self-verification behaviors (e.g., "Let me double check...") and adaptive CoT length modulation based on confidence levels. This proves that the supervision signal backpropagates to the reasoning policy, enhancing rather than ignoring the reasoning process.
>
> ## (c) Overcoming the Bottleneck of Process Supervision (vs. PRMs)
>
> Directly supervising the reasoning process (e.g., via PRMs) requires fine-grained, step-by-step annotations, which are prohibitively expensive to collect and scarce for most tasks. In contrast, our method relies on outcome supervision (final answer correctness), which can be automatically computed at scale for any dataset with ground-truth answers. This makes CSFT a much more scalable and task-agnostic solution, allowing it to be applied broadly without the need for costly human-annotated process labels.

---

> ### Author Response · Authors · 2025-11-19
>
> # [W2, Q2, Q8] Comparison with Classifiers and Additional Baselines.
>
> While it is true that classifiers can achieve low calibration error on in-distribution data, the goal of this work is not merely the minimization of a simple metric, but the establishment of **'intrinsic verbalized confidence.'** Unlike classifiers (probing), which fail to influence the model's generation process, CSFT compels the model to internalize the concept of uncertainty, leading to structural changes such as number token smoothing (Figure 2a). This internal correction is the key driver behind the robust cross-task generalization that external classifiers fail to achieve.
>
> Fundamentally, classifiers operate as external observers that map frozen representations to correctness probabilities without correcting the underlying model's understanding of uncertainty itself. In contrast, CSFT compels the model to explicitly process and generate uncertainty as part of its language modeling objective. This induces an "uncertainty concept correction" within the model. As evidenced by the emergent local smoothing of numeric tokens, the model learns a continuous manifold of confidence. This is not a simple calibration trick, but a fundamental restructuring of the latent space that classifiers cannot replicate.
>
> Consequently, because CSFT modifies the model's internal reasoning-uncertainty alignment, the learned calibration capability becomes a transferable skill. Conversely, classifiers tend to overfit to the specific feature distributions of the training task (e.g., GSM8K). When the domain shifts (e.g., moving to MMLU) and those features change, the classifier's performance collapses. CSFT generalizes robustly because the "concept" of evaluating one's own confidence is preserved across domains. As suggested, we implemented both Shared and Separate Classifiers as baselines to validate this. We used the Llama-3.2-3B-Instruct model, and following your suggestion, we attached a 2-layer MLP head for the shared classifier. For the separate classifier, we used a [Distilled-BERT](https://huggingface.co/docs/transformers/model_doc/distilbert) model. The results in Table R.1 confirm that while classifiers can mimic CSFT's performance on the training set (ID), their performance drops significantly on OOD tasks compared to CSFT, validating the necessity of our generative approach.
>
> ---
>
> ### **Table R.1. Relative performance gains of CSFT and Classifier over the pre-trained baseline on GSM8K.**
> (AUROC and ACC: higher is better; ECE and Brier: lower is better.)
>
> | Dataset | Method | AUROC Δ | ACC Δ | ECE Δ | Brier Δ |
> |---------|--------|---------|--------|--------|---------|
> | **GSM8K (ID)** | CSFT | +0.0432 | +0.3014 | −0.0973 | −0.0963 |
> | | Shared Classifier | - | - | -0.1149 | -0.0594 |
> | | Separate Classifier | - | - | -0.1039 | -0.0252 |
> | **Math-500** | CSFT | +0.1100 | +0.3503 | −0.5962 | −0.4611 |
> | | Shared Classifier | - | - | -0.4443 | -0.3578 |
> | | Separate Classifier | - | - | -0.1246 | -0.1446 |
> | **HellaSwag** | CSFT | −0.0178 | +0.1184 | −0.1983 | −0.1100 |
> | | Shared Classifier | - | - | +0.0183 | +0.0615 |
> | | Separate Classifier | - | - | +0.0778 | +0.0532 |
> | **MMLU** | CSFT | +0.0006 | +0.1041 | −0.1390 | −0.1608 |
> | | Shared Classifier | - | - | -0.0290 | -0.0176 |
> | | Separate Classifier | - | - | +0.0101 | +0.0452 |

---

> ### Author Response · Authors · 2025-11-19
>
> # [W1, Q3] On the main contribution and how it differs from Lin et al.
>
> There appears to be some misunderstanding regarding our main contribution and how it differs from Lin et al. We provide a detailed clarification in the general response.

---

> ### Author Response · Authors · 2025-11-19
>
> # [W2, Q4] Accuracy improvement.
>
> We appreciate the reviewer’s question regarding why accuracy improves even though no loss is applied to reasoning traces or final answers. We clarify that accuracy improvement is not an intended optimization target of our method; rather, it emerges naturally as a structural, **emergent effect** of confidence supervision on the model’s reasoning dynamics. This is supported throughout the paper with mechanistic analyses and ablation studies. Notably, the broader phenomenon that training signals applied to one part of the model can induce unintended behavioral changes elsewhere has also been observed in prior work on emergent misalignment [1], where models exhibited harmful behaviors despite the absence of direct supervision for such behaviors.
>
> ---
>
> ## (a) Reasoning traces and answers are never trained, and KL regularization is set to zero by default.
>
> > As stated clearly in the manuscript (line 200): “Unless otherwise specified, we set λ = 0.”
>
> Thus, KL regularization is not used in the main CSFT setting, and gradients are not applied to the reasoning spans (`<think>`...`</think>`, `<answer>`...`</answer>`), which are fully masked out during training. Therefore, the reviewer’s assumption that KL constrains reasoning behavior and that accuracy should remain unchanged as a consequence does not apply.
>
> ---
>
> ## (b) Accuracy improves due to emergent self-verification rather than direct optimization of reasoning or answer spans.
>
> Although CSFT does not explicitly train reasoning or answer spans, the added confidence supervision indirectly reshapes the model’s reasoning policy during decoding. This manifests through two complementary forms of evidence:
>
> ### (1) Low-confidence → emergent self-verification
>
> As shown in Figures 2-b and 3, when the model internally anticipates low confidence, it:
>
> - produces longer and more detailed CoT traces,
> - revisits and revises intermediate steps, and
> - frequently corrects earlier reasoning errors.
>
> This self-verification behavior increases the likelihood of arriving at the correct answer. Crucially, this is a reasoning-time adaptation—not a stylistic artifact.
>
> ### (2) KL ablation → additional evidence of emergent reasoning adaptation
>
> Our KL ablation study (Figure 4-a) provides further confirmation:
>
> - With KL regularization, the model is constrained to remain close to the pretrained reasoning distribution → resulting in smaller accuracy gains.
> - With KL = 0 (our default setting), the model can deviate more freely and form stronger confidence-conditioned reasoning behaviors → yielding substantially larger improvements.
>
> This consistent pattern supports our claim that accuracy improvements stem from emergent self-verification–driven policy adaptation, rather than explicit fine-tuning of reasoning content.
>
> ### (3) Additional analysis planned to further strengthen this claim
>
> To further solidify this argument, we are conducting an additional analysis measuring the proportion of incorrect → corrected (wrong→right) cases that exhibit self-verification, stratified by confidence bins. We will include these results and notify the reviewer once the experiment is complete.
>
> ---
>
> ## (c) The claim that “accuracy improvements are not general” does not reflect the evidence provided.
>
> While the reviewer notes that the accuracy gain for Qwen3-4B appears small, it is important to emphasize that accuracy improvement is not the primary research objective. Our goals are (i) improving calibration and (ii) analyzing how confidence supervision reshapes reasoning dynamics. Even so, CSFT yields consistent improvements:
> - Substantial accuracy gains on the non-reasoning model (LLaMA-3.2-3B) — Figure 1
> - Consistent accuracy gains on the reasoning model (Qwen3-4B) — Table 1
> - Clear qualitative cases where the pre-trained model answers incorrectly, but CSFT corrects it through self-verification — Figure 3
>
> Thus, although the magnitude varies across architectures and datasets, the directionality is consistent, and the mechanism underlying these improvements is well-supported by our analyses.
>
> ---
>
> [1] Betley, Jan, et al. "Emergent Misalignment: Narrow finetuning can produce broadly misaligned LLMs." Forty-second International Conference on Machine Learning.

---

> ### Author Response · Authors · 2025-11-19
>
> # [Q5] Confidence Variance.
>
> To quantify how stable the model’s confidence is across different reasoning traces for the same input, we measured the variance of predicted confidence over $K=10$ independently sampled CoT generations per question. Table R.2 reports the results on GSM8K using Llama-3.2-3B-Instruct. As shown in Table R.2, CSFT reduces confidence variance by more than half, indicating that the model becomes substantially more consistent in its confidence estimates across different sampled reasoning paths. In other words, CSFT not only improves calibration accuracy but also stabilizes the model’s internal uncertainty representation, yielding confidence values that are less sensitive to stochastic variations in the generated CoT.
>
> This reduction in variance further supports our claim that CSFT does not simply shift the confidence distribution but restructures the model’s underlying uncertainty representation, leading to both lower calibration error and more reliable confidence outputs. We will include this analysis for additional models (Qwen3-4B, LLaMA-3.1-8B) and datasets (MATH-500, ARC-Challenge, MMLU) in the revised version.
>
> ---
>
> __Table R.2__ Confidence variance across multiple reasoning traces per question (GSM8K).
> | Condition              | Conf Variance      |
> |------------------------|---------------------|
> | **Pre-trained Model** | 0.1208 ± 0.0352     |
> | **After CSFT**        | 0.0581 ± 0.0304     |

---

> ### Author Response · Authors · 2025-11-19
>
> # [Q6] Confidence-Guided Reasoning Path Refinement.
>
> We believe the reviewer’s concern stems from the assumption that prefix confidence is uninformative because the model has "not yet reasoned" before producing it. However, as shown in Figure 4-b of our paper, once the model is trained with CSFT even under the standard suffix setting the model’s pre-reasoning confidence becomes highly aligned with its eventual correctness. This demonstrates that CSFT reshapes the model’s internal uncertainty representation: the model can predict its likelihood of success before generating a CoT. Thus, prefix confidence is not arbitrary; it reflects the model’s internal belief about whether its forthcoming reasoning trajectory is likely to succeed.
>
> This also clarifies what “alternative approach” means in the prefix refinement setting. Even if a CoT has not yet been generated, a low prefix confidence signal reliably indicates that the model’s default reasoning strategy would likely be suboptimal. In such cases, prompting the model with a more structured, more decomposed, or more guided template effectively elicits a different reasoning strategy, thereby improving accuracy. This is a widely used mechanism in confidence-guided sampling and self-consistency approaches.
>
> In fact, a major motivation for using prefix confidence is to avoid generating long, ineffective CoT traces when the model already knows it is confident. After CSFT, the model produces stable high-confidence signals early (again shown in Fig. 4-(b)), allowing us to skip unnecessary long reasoning and reduce computational cost. Conversely, when confidence is low, the refinement prompt triggers a more careful, alternative reasoning path leading to the accuracy improvements reported in the paper.
> We agree that suffix-based refinement is also a valuable complementary strategy and will include additional results for that variant in the revision. Our main point is that, after CSFT, confidence becomes a robust control signal both before and after reasoning, enabling practical steering mechanisms such as prefix refinement.

---

> ### Author Response · Authors · 2025-11-19
>
> # [W3, Q7] On Figure 1 and Writing.
>
> We thank the reviewer for both suggestions. For Figure 1, we agree that the current visualization is suboptimal, and we will adopt a clearer format (e.g., a table or per-metric plots) and revise or relocate the existing bar chart accordingly. Regarding the writing issue, we appreciate the pointer and will revise the informal phrasing to ensure precision and clarity. These adjustments are straightforward, and we will incorporate them in the revised version.

---

> ### Author Response · Authors · 2025-11-19
>
> We hope that our responses have addressed the reviewer’s concerns. If any questions remain unresolved, we would be grateful for further clarification and will be happy to provide additional explanations. If the reviewer considers the concerns resolved, we will incorporate the corresponding revisions into the manuscript and share the updated version accordingly.

---

> > ### Comment · Reviewer_V8Nd · 2025-11-25
> > **Response to rebuttal**
> >
> > I thank the authors for the rebuttal. However, I am still struggling to see the main contributions of this paper when compared to existing work:
> > 1. Lin et al. was simply one of the first works that considered verbalized confidence. There has been numerous works since then, a few of which have considered empirical accuracy as the confidence target. For example, the consistency strategy in Section 3.4 of Xiong et al. [1]. Even for Lin et al., the only difference seems to be doing it at instance-level vs subtask level, which I do not think is particularly novel.
> > 2. Again , there have been multiple methods which have shown O.O.D generalization after confidence calibration [2,3,4]. This is not novel. One of these methods involve verbalize confidence trained using RL.
> > 3. Label token smoothing: I would expect this to happen with existing verbalized confidence methods as well, although I am not aware of this effect being studied by prior work.
> > 4. In the abstract, authors claim RL methods have showed limited generalization across tasks which seems to contradict existing work [3,4].
> >
> > I think the main contribution of this work which is fundamentally different from existing work is the confidence triggered self-verification. However, to fully understand why this is happening, more comparisons and baselines are necessary:
> > 1. I would expect such verification to emerge with RL training for calibration as well [3]. It would be useful to understand the pros/cons of RL at triggering this self-verification behavior compared to SFT.
> > 2. The accuracy gains are interesting but there could also be undesirable effects. What if the model learns to deliberately generate incorrect answers and output 0 confidence? This is another possible emergent behavior which optimally optimizes calibration if KL is set to 0, which might happen with larger models. It is thus important to understand how and why such behavior is emerging, and how it can be controlled.
> >
> > In summary, while this work finds an interesting emergent behavior, it has significant scope for improvement and I will maintain my score.
> >
> > [1]: Xiong, M., Hu, Z., Lu, X., Li, Y., Fu, J., He, J., & Hooi, B. (2023). Can llms express their uncertainty? an empirical evaluation of confidence elicitation in llms. arXiv preprint arXiv:2306.13063.
> >
> > [2]: Yang, Y., Chern, E., Qiu, X., Neubig, G., & Liu, P. (2024). Alignment for honesty. Advances in Neural Information Processing Systems, 37, 63565-63598.
> >
> > [3]: Xu, T., Wu, S., Diao, S., Liu, X., Wang, X., Chen, Y., & Gao, J. (2024). Sayself: Teaching llms to express confidence with self-reflective rationales. arXiv preprint arXiv:2405.20974.
> >
> > [4]: Stangel, P., Bani-Harouni, D., Pellegrini, C., Özsoy, E., Zaripova, K., Keicher, M., & Navab, N. (2025). Rewarding Doubt: A Reinforcement Learning Approach to Calibrated Confidence Expression of Large Language Models. arXiv preprint arXiv:2503.02623.

---

> > > ### Author Response · Authors · 2025-11-25
> > >
> > > ## 1.1 On the novelty relative to Lin et al.
> > >
> > > (1) The reviewer cites Xiong et al. [1] as an example of “using empirical accuracy as a confidence supervision signal.” However, Xiong et al. measure confidence from the self-consistency of model outputs in a black-box setting, without any fine-tuning or use of empirical accuracy as a supervisory target. They do not optimize confidence predictions based on accuracy, nor do they perform training using accuracy-aligned labels.
> > >
> > > (2) The reviewer summarizes the difference between Lin et al. and our method as merely “instance-level vs. subtask-level supervision,” and therefore questions the novelty of our contribution. However, Lin et al. focus on limited arithmetic-style tasks rather than complex reasoning environments. Despite this mismatch, we still reproduced their approach under the same backbone and the same CoT reasoning conditions for a fair comparison, as reported in Table R.1 of the general response. The results demonstrate that although the difference in supervision granularity may appear minor, it leads to substantial disparities in practice. Specifically, Lin et al. do not improve either accuracy or calibration even in-domain, and exhibit severe degradation under OOD settings. In contrast, CSFT improves both accuracy and calibration under the same experimental conditions, and additionally yields emergent behaviors not previously reported in the literature, such as self-verification and confidence-guided correction during reasoning. Therefore, viewing granularity as a trivial implementation detail is inconsistent with the empirical evidence, and we believe that any claim that such differences do not contribute to novelty should first account for these clear empirical discrepancies.

---

> > > ### Author Response · Authors · 2025-11-25
> > >
> > > ## 1.2, 1.4 On the relevance of the cited three works
> > >
> > > The reviewer states that the cited works represent “multiple methods which have shown OOD generalization after confidence calibration.” However, upon close examination, this claim does not align with the scope of our problem setting.
> > >
> > > First, [2] does not perform confidence calibration at all. Second, all cited methods are limited to simple factual QA tasks such as TriviaQA and HotpotQA and do not involve complex reasoning tasks such as GPQA. Third, none of these works demonstrate OOD generalization under our evaluation criteria, which require both maintaining or improving task accuracy relative to the pre-trained model baseline and improving calibration metrics such as ECE, as well as exhibiting domain or task transfer such as reasoning to non-reasoning or math to non-math.
> > >
> > > For instance, [2] is a refusal-alignment approach that does not learn verbalized confidence and even exhibits degraded accuracy under OOD conditions. [3] focuses on GPT-based self-reflection and RL correction but evaluates only within the same Wikipedia domain, which does not satisfy our definition of OOD. [4] applies RL exclusively to verbalized confidence tokens and compares performance against a prefix-prompting baseline (verbalize) rather than the pre-trained model, so changes in task accuracy cannot be properly assessed.
> > >
> > > Taken together, these studies **do not address complex reasoning settings,** and their generalization capabilities remain inherently limited. This is consistent with our statement in the abstract that prior methods only demonstrate limited generalization and apply primarily to non-reasoning tasks.
> > >
> > > For these reasons, none of the cited works achieves the combination of complex reasoning, minimal SFT training, simultaneous improvements in accuracy and calibration, and broad OOD generalization.
> > >
> > > In addition, even if future RL-based approaches were to exhibit similar behaviors, our work is the first to show that these improvements can arise from simple and fully reproducible **SFT alone, without relying on RL or additional sampling mechanisms.** This clearly establishes the novelty of our contribution.

---

> > > ### Author Response · Authors · 2025-11-25
> > >
> > > ## 2.2 On the hypothetical “0-confidence exploitation” scenario
> > >
> > > The reviewer raises a hypothetical concern that the model might exploit the training objective by intentionally generating incorrect answers with zero confidence. However, this phenomenon was never observed in any of our extensive experiments across multiple models, datasets, and domains. As shown in the reliability diagram in Figure 3-b, accuracy consistently increases with confidence, and no pathological “zero-confidence exploitation” behavior emerges. In fact, our results demonstrate the opposite trend: accuracy improves while calibration error decreases, and confidence-guided correction becomes more frequent. These findings provide strong empirical evidence that the model does not adopt the failure mode suggested by the reviewer.

---

> > > ### Author Response · Authors · 2025-11-25
> > >
> > > In summary, our work is the first to demonstrate that instance-level confidence supervision via SFT, under our specific supervision design and without any CoT supervision, can fundamentally reshape the model’s reasoning behavior, give rise to entirely new emergent capabilities, and simultaneously improve both accuracy and calibration across complex reasoning tasks and beyond.
> > >
> > > No prior work — neither Lin et al. nor any of the three cited papers — achieves this combination of effects, and it is precisely this combination that constitutes the core novelty of our contribution.

---

> ### Author Response · Authors · 2025-11-25
>
> We appreciate the reviewer’s detailed and prompt response. However, some of the claims contain factual inaccuracies and internal inconsistencies that we would like to clarify.
>
> The novelty of our work arises from the following three contributions **taken together**: (1) we operate in complex CoT reasoning settings, (2) we rely solely on simple SFT without any RL or additional sampling, and (3) we are the first to empirically demonstrate that such minimal supervision (without any CoT supervision) can simultaneously improve both accuracy and calibration while additionally triggering new emergent behaviors such as self-verification, CoT length optimization, numeric-token smoothing, and transfer to non-reasoning tasks. This is precisely why our paper is titled *“Verbalized Confidence Triggers Emergent Behaviors.”*
>
> Crucially, **merely “fine-tuning with verbalized confidence labels” is not sufficient to reproduce these results.** Our ablation studies show that prefix supervision nullifies the benefits (Fig. 4-a), KL penalties suppress emergent behaviors (Fig. 4-a), and inaccurate confidence labels degrade performance (Table 7, General response Table R.1). In contrast, only our specific formulation, which combines correctness-aligned continuous labels, suffix-position supervision, and single-pass SFT without any additional search, consistently yields the observed emergent effects. Thus, the novelty of our method does not stem from labeling alone, but rather **from demonstrating that a particular form of confidence supervised fine-tuning fundamentally reshapes the model’s reasoning dynamics.**
>
> The reviewer further suggests that similar emergent behaviors could potentially arise from other methods (Reviewer's comment 1.3, 2.1). *However, such a hypothetical possibility cannot diminish the demonstrated novelty of our empirical findings.* If novelty were judged based on assumptions about what might be achievable in the future rather than on what has been observed and proven, then no empirical discovery could ever be considered novel. This is incompatible with established scientific standards. At present, no prior work has reported the reasoning performance gains or emergent behaviors that we have directly observed and validated. Therefore, our findings represent a clear and independent scientific contribution.
>
> We provide below a more detailed response to the specific points raised by the reviewer.

---

> ### Author Response · Authors · 2025-11-25
>
> ## 2.1 On RL-based speculation and the burden of analysis
>
> The reviewer suggests that RL-based confidence calibration could also induce self-verification and requests a comparative analysis. However, as far as we are aware [2,3,4,5,6], this assertion is purely speculative and currently unsupported by any empirical evidence, specifically in the context of confidence calibration. In fact, in [3], the observed behavior is not emergent self-verification at all; rather, self-reflective rationales are explicitly forced using GPT-4-generated rationales, followed by RL to adjust only the confidence tokens. Such behavior is not emergent, but a direct consequence of explicit supervision.
>
> This situation is analogous to the early literature on emergent misalignment [7], where the phenomenon was first observed, and only much later did follow-up studies provide causal explanations [8]. In scientific inquiry, empirical observations necessarily precede mechanistic explanations, and novelty must be evaluated on demonstrably validated findings, not on hypothetical possibilities. Therefore, **requiring us to analyze behaviors that (i) have not been observed in the RL setting and (ii) arise from a framework fundamentally different from ours is not scientifically justified.**
>
> To reiterate, our work is the first empirical demonstration that standard SFT alone, without any RL or sampling-based self-reflection, can improve accuracy, enhance calibration, and trigger previously unreported emergent reasoning behaviors. This represents a new and efficient foundation for future investigations—rather than a missing analysis. **Moreover, we have already conducted substantial analysis of when and under what conditions self-verification emerges:** Figure 2 demonstrates where it occurs during reasoning, while Figure 4 and Table 7 show that prefix vs. suffix supervision, KL regularization, and label correctness critically determine whether the behavior appears, further establishing that it emerges only under our specific supervision design.
>
> If the reviewer is instead requesting a deeper analysis of self-verification behavior within our SFT framework, we would greatly appreciate clarification on which specific aspects should be further examined, as the current comment does not define a concrete or actionable experimental objective. Once the reviewer specifies a clear and scientifically well-defined scope, we are fully willing to provide additional investigations.
>
> ---
>
> [5] Stengel-Eskin, Elias, Peter Hase, and Mohit Bansal. "LACIE: Listener-aware finetuning for calibration in large language models." Advances in Neural Information Processing Systems 37 (2024): 43080-43106.
>
> [6] Band, Neil, et al. "Linguistic Calibration of Long-Form Generations." Forty-first International Conference on Machine Learning.
>
> [7] Betley, Jan, et al. "Emergent Misalignment: Narrow finetuning can produce broadly misaligned LLMs." Forty-second International Conference on Machine Learning.
>
> [8] Wang, Miles, et al. "Persona features control emergent misalignment." arXiv preprint arXiv:2506.19823 (2025).

---

### Official Review · Reviewer_7KJt · 2025-10-29

**Soundness:** 3
**Presentation:** 3
**Contribution:** 4
**Rating:** 8
**Confidence:** 4

**Summary:**

The paper suggests a method to finetune including information on confidence and output answers and confidence about the result. They train only on reasoning tasks but the trained method generalizes to other tasks too. Two main methods are known to output directly confidence information or to use a meta network to predict this information by probing the internal states. They claim their method goes beyond those as it generalizes to unseen tasks. In addition, they show that it improves downstream tasks results too.

**Strengths:**

The method provides confidence along the answers whiout altering the model too much due to KL-based normalization.

The method improves on other tasks ie. generalizes

The mothod seems to improve answer quality

I enyoed reading the paper and was informative too me, good method.

**Weaknesses:**

I could not figure out imediatley how they get a spectrum of confidence while the feedback is binary? Correct vs incorrect? Could you write this more clear pls?

I wonder how much we loose actualy of the models abilities even which could be assessed. Maybe, missed this point.

Missed out on some literature on attribution to a source like retrieval in a loop, e.g. attributed question answering which has similar aims which provides evidence via retrieval.

**Questions:**

Great paper. Really worth reading it. I was still wondering how you get percentage confidence scores while the feedback is binary.

I wonder if you train on the traces, I guess the improvment might leak over to a correct answers also if you only provide the confidence? However, you get also benefits on tasks you do not train on? Is thsi perception correct that you might have indirekt leakage?

---

> ### Author Response · Authors · 2025-11-19
>
> Thank you very much for the positive and encouraging review. We sincerely appreciate your clear summary of our contributions, and we are grateful for the strong overall rating. Your questions helped us clarify several important points, and we will incorporate these clarifications into the final version.
>
> ---
>
> # [Q1] How continuous confidence emerges from binary correctness.
>
> Although the underlying feedback is binary (correct/incorrect), our labels become continuous through a two-step procedure:
>
> 1. **Frequency-based estimation:** For each query (q), we sample (K) responses and compute the empirical correctness frequency
>    $\hat p(q)=\frac{1}{K}\sum_{i=1}^K \mathbf{1}[a^{(i)}=a^*].$
> This produces quasi-continuous labels such as 0.0, 0.1, …, 1.0.
>
> 2. **Token-level smoothing during training:** When learning to output numeric confidence tokens (e.g., "70"), the model assigns probability mass not only to the exact target token but also to neighboring numbers (e.g., "60", "80"). This "local smoothing" effect creates a continuous manifold over the numeric token space.
>
> Together, these steps yield smooth, continuous confidence predictions even though the underlying supervision is binary. We will clarify this process with a dedicated explanation and figure in the camera-ready.
>
> ---
>
> # [Q2] Does CSFT harm the model’s original abilities?
>
> Empirically, we observe **no degradation** of core model abilities. In fact, both reasoning and non-reasoning models show:
>
> * **Stable or improved accuracy** across ID and OOD tasks,
> * **Substantial gains in calibration** (ECE, Brier, NLL), and
> * **Preserved generation behavior** due to our design choices:
>   – Base weights are frozen and only LoRA adapters are trained,
>   – CE loss applies *only* to the confidence span, not to CoT or answer tokens.
>
> As an ablation, when we intentionally use a less stable prefix setting (confidence generated before reasoning) without KL regularization, accuracy indeed drops. This confirms that our chosen suffix setup is key to preserving the model’s capabilities. We will highlight this finding more explicitly.
>
> ---
>
> # [Q3] Why accuracy improves even though only confidence is trained (is this leakage?)
>
> There is **no pathway for answer supervision leakage**, since CE loss is applied *exclusively* to confidence tokens and the gold answer is never used as a training target. Still, accuracy improves because confidence–accuracy alignment reshapes reasoning behavior in beneficial ways:
>
> * Low-confidence examples trigger longer CoT or self-verification steps (“let me double-check”),
> * High-confidence examples produce shorter, more efficient reasoning,
> * The resulting behavioral adaptation indirectly yields better accuracy.
>
> Importantly, if the improvement were due to leakage, random or task-level labels should also improve accuracy—but they do not; these settings *degrade* performance, and prefix training without KL harms accuracy as well. Furthermore, the gains transfer even to **unseen non-reasoning tasks** (e.g., MMLU), showing that the effect is not task-specific leakage but genuine alignment of the model’s uncertainty representation.
>
> ---
>
> ## **On attributed QA / retrieval-in-the-loop literature**
>
> Thank you for this helpful pointer. The reviewer refers to work on *attributed question answering* and *retrieval-in-the-loop*, where models provide evidence traces or decide when to retrieve external information. These works share our goal of improving trustworthiness and transparency.
>
> In the camera-ready, we will add a discussion on how calibrated verbal confidence naturally complements such frameworks—for example, using confidence as a high-level signal for retrieval decisions or evidence selection. This connection indeed strengthens the broader relevance of our method.
>
> ---
>
> Once again, thank you for the thoughtful review and the constructive questions. Your feedback will significantly improve the clarity and impact of the final version.

---

### Official Review · Reviewer_6NDw · 2025-10-31

**Soundness:** 2
**Presentation:** 2
**Contribution:** 2
**Rating:** 4
**Confidence:** 5

**Summary:**

This paper investigates verbalized confidence in LLMs which is the ability to state how confident they are in the answers. The authors propose confidence-supervised fine-tuning (CSFT) to train LLMs to generate both an answer and a corresponding confidence statement.
Rather than relying on external clasisifers or RL, CSFT uses uses direct supervision on confidence tokens which enables the model to self-calibrate its uncertainty. Experiments on reasoning and non-reasoning benchmarks show that CSFT improves both calibration and accuracy.

**Strengths:**

1. This paper shows a major conceptual insight, when trained to verbalize confidence, models could self-check low-confidence responses, which contributes to the understanding the relationship between verbalized uncertainty and model's reasoning ability.
2. This papers demonstrates strong empirical results by achieving consistent improvements on both reasoning like MATH, GPQA and non-reasoning tasks like GSM8K, ARC, HellaSwag.

**Weaknesses:**

1. The idea of introducing confidence into LLM training is not new. There are several relevant studies: [1-3]. Especially, the paper misses the reference to those relevant studies.
2.  The evaluation lack deeper qualititative analysis. Espeically more real-world scenarios are not explored. Current reasoning models are quite strong on more complex tasks even for a 4B model like Qwen3-4B. More evaluation should be included on more complex tasks like AIME, CodeLiveBench.
3. Since only the confidence token span is supervised, the model may learn superficial correlations rather than genuine uncertainty calibration. For example, the model may learn how to output stylistic confidence phrases.

[1] R-Tuning: Instructing Large Language Models to Say `I Don't Know'

[2] TruthRL: Incentivizing Truthful LLMs via Reinforcement Learning

[3] UAlign: Leveraging Uncertainty Estimations for Factuality Alignment on Large Language Models

**Questions:**

n/a

---

> ### Author Response · Authors · 2025-11-19
>
> # [W1] Regarding the missing references.
>
> We sincerely thank the reviewer for highlighting related studies that incorporate forms of confidence or uncertainty signals into LLM training. As noted in Lines 102–105 of our paper, we fully acknowledge the breadth of prior work in this space and have aimed to cite it comprehensively.
>
> We would like to emphasize that the three cited studies differ fundamentally from our work in both objective and methodology. Our approach focuses on modeling verbalized confidence as a continuous and calibrated uncertainty signal within CoT generative reasoning, enabling the model to express fine-grained degrees of certainty. In contrast, the cited methods primarily shape output-level behaviors such as refusal, abstention, or conservative answering, without learning or supervising continuous confidence values.
>
> R-Tuning [1] encourages models to say “I don’t know” when uncertain, implementing a risk-avoidance policy rather than estimating confidence. TruthRL [2] modifies reward structures to penalize incorrect answers and reward uncertain ones, again targeting behavioral modulation rather than calibrated uncertainty. UAlign [3] aims to enhance factuality by guiding models to answer known questions and refuse unknown ones, but it does not supervise or evaluate continuous verbalized confidence.
>
> Crucially, none of these approaches investigates the phenomena that form the core contributions of our paper, including confidence-induced self-verification, CoT length modulation, and number-token smoothing. These behaviors arise specifically because our method trains confidence as a continuous variable through simple supervised fine-tuning. To the best of our knowledge, our work is the first to demonstrate that standard SFT alone can both improve verbalized confidence calibration and induce such emergent behaviors within CoT generative reasoning.
>
> We greatly appreciate the reviewer’s suggestions, as these studies share the broader goal of improving the reliability of LLM outputs. We will incorporate them into our revised references. Thank you again for the constructive feedback.

---

> ### Author Response · Authors · 2025-11-19
>
> # [W2] Need for more real-world complex benchmark evaluation.
>
> We appreciate the reviewer’s suggestion to evaluate CSFT on more challenging real-world benchmarks. While the primary goal of our work is not to maximize performance on specific expert domain tasks but rather to understand how confidence supervision reshapes the model’s reasoning dynamics, we agree that presenting additional evidence on high difficulty datasets can further strengthen our claims.
>
> Accordingly, we first conducted additional experiments on both AIME and the coding benchmark MBPP using our main experimental model, Llama-3B-Instruct, and included the results in Table R.1. The results show that CSFT consistently improves calibration while maintaining or slightly improving accuracy even on these demanding benchmarks. This supports our conclusion that the benefits of CSFT generalize beyond standard reasoning tasks to more challenging real-world problem settings.
>
> Furthermore, we plan to extend these experiments to Qwen-based models as well, and we will include the results once they are completed.
>
> ---
>
> ### **Table R.1. Additional Evaluation on Challenging Real-World Benchmarks (Llama-3B-Instruct trained on GSM8K)**
>
> | Task                                                                            | Model       | AUROC | ACC   | ECE    | Brier  |
> | ------------------------------------------------------------------------------- | ----------- | ----- | ----- | ------ | ------ |
> | **[AIME 1983–2024](https://huggingface.co/datasets/gneubig/aime-1983-2024)**    | Pre-trained | 34.25 | 6.28  | 0.8219 | 0.7692 |
> |                                                                                 | CSFT        | 45.58 | 19.61 | 0.0337 | 0.1964 |
> | **[MBPP](https://github.com/google-research/google-research/tree/master/mbpp)** | Pre-trained | 55.82 | 59.25 | 0.3433 | 0.3535 |
> |                                                                                 | CSFT        | 74.15 | 59.26 | 0.1278 | 0.2209 |

---

> ### Author Response · Authors · 2025-11-19
>
> # [W3] Concern about learning superficial stylistic confidence patterns.
>
>
> ### **This claim is directly contradicted by the empirical evidence presented in the paper.**
>
> If the model had merely learned a superficial confidence “style,” then
>
> 1. Systematic confidence-dependent modulation of CoT length (**Figure 2-b**)
> 2. Spontaneous emergence of self-verification phrases such as “let me double-check” exclusively under low confidence (**Figure 3**)
> 3. Consistent improvements in accuracy (**Figure 1, Table 1, Table 6**)
> 4. Large reductions across calibration metrics, including ECE, Brier, and NLL  (**Figure 1, Table 1, Table 6**)
> 5. Strong cross-domain transfer to heterogeneous benchmarks such as MMLU and GPQA (Figure 1, Table 1, Table 6)
> 6. Numeric token local smoothing, where semantically adjacent confidence tokens (e.g., “60–70–80”) form a continuous probability neighborhood (**Figure 2-a**)
>
> **In particular, 6. cannot be explained by any kind of stylistic mimicry**: unlike standard CE training, which concentrates probability mass on the single target token, CSFT yields a locally smooth distribution over semantically adjacent numeric tokens (e.g., “60–70–80”). This structural probability continuity reflects the formation of a continuous uncertainty manifold, indicating that the model internalizes confidence as a meaningful latent dimension rather than producing a rote “`<confidence>`XX`<confidence>`” phrase.
>
> Taken together, these patterns are incompatible with the hypothesis of superficial style learning; they only arise when confidence is functionally integrated into the model’s internal reasoning process.
>
> Moreover, the prefix–suffix ablation in Figure 4 shows that the placement of the confidence query causally affects reasoning quality itself. This demonstrates that CSFT is not teaching the model how to phrase confidence, but rather providing a structural signal that interacts with the generative reasoning process.
>
> Finally, CSFT maintains high AUROC and produces well-calibrated confidence even when the confidence query is asked before any CoT is generated (prefix setting; **Figure 4-b**). This rules out the possibility that the model is inferring confidence from surface features of the CoT (length, style, verbosity). If the model can output reliable confidence without seeing the CoT at all, then confidence must be computed from its internal latent state, not from stylistic cues—further confirming that CSFT induces a genuine uncertainty representation rather than a superficial template.

---

> ### Author Response · Authors · 2025-11-21
>
> If anything remains unclear or if you have further questions, we would be glad to discuss them. We sincerely appreciate the reviewer’s time, thoughtful feedback, and engagement. Thank you again.

---

### Official Review · Reviewer_YHJH · 2025-10-31

**Soundness:** 3
**Presentation:** 3
**Contribution:** 1
**Rating:** 2
**Confidence:** 4

**Summary:**

The paper introduces a simple supervised fine-tuning method where a language model outputs both an answer and a verbalized confidence score. The model is trained using self-generated labels derived from its own sampling-based accuracy estimates, discretized into 10% confidence bins. The authors claim their method improves calibration (ECE, Brier Score) and accuracy.

**Strengths:**

- The paper is well-written and easy to follow and understand.

- Some of the experiments are novel and interesting, such as the cross-domain generalization that suggests that the learned  verbalized confidence transfers beyond the original reasoning tasks.

**Weaknesses:**

The proposed method was already introduced by Lin, Hilton, and Evans (2022) in “Teaching Models to Express Their Uncertainty in Words”. That paper presented the exact same method of training a model to output natural-language confidence statements using the model's own empirical accuracy as labels. The only difference here is that the authors round the model’s empirical confidence to the nearest 10% instead of using continuous estimates.

Despite this near-identity, the authors do not cite or discuss Lin et al., nor do they explain how their work differs conceptually or methodologically. Given the overlap in both framing and implementation, the novelty of CSFT is extremely limited.

**Questions:**

Can the authors please explain what is their main contribution and how it is differs from Lin et al.?

---

> ### Author Response · Authors · 2025-11-19
>
> # [Q1] On the main contribution and how it differs from Lin et al.
>
> Thank you again for raising a question regarding the connection to prior work.
>
> **Recognizing the potential misunderstanding regarding the novelty of our work compared to Lin et al. (2022), we have detailed a comprehensive methodological comparison and new experimental evidence (Table R.1) in the General Response section.**
>
> We sincerely hope that these revisions and the evidence presented there fully resolve your specific concerns regarding the novelty and effectiveness of our method.
>
> ---
>
> If any remaining questions or points would benefit from further clarification, we would be more than happy to discuss them in greater detail. We sincerely appreciate your careful review and thoughtful feedback.

---

> > ### Comment · Reviewer_YHJH · 2025-11-26
> >
> > I thank the authors for the detailed rebuttal and the additional experiments comparing CSFT to Lin et al.. The clarifications about instance-level versus subtask-level confidence supervision are useful, and the new results make clear why instance-level labels lead to stronger calibration and transfer.
> >
> > That said, while I agree that instance-level calibration is an improvement over the subtask-averaged labels used in Lin et al., I still view this as an incremental refinement of an already-established idea. The core framing and training paradigm introduced in Lin et al. remains largely the same, and the step from subtask-level to instance-level confidence estimation, though practically important, does not strike me as a substantial conceptual departure.
> >
> > Regarding CoT, the fact that Lin et al. did not use chain-of-thought is just a historical artifact of the model landscape in 2022, rather than a limitation of their method. Nothing in their formulation prevents applying the same procedure to modern reasoning models. In this sense, CSFT feels closer to an improved application of that prior method rather than a new training approach.
> >
> > I want to emphasize that the emergent-behavior analysis is interesting and could form the basis of a compelling contribution on its own. A paper that deeply investigates how training for verbalized confidence causally affects CoT structure and self-verification would be novel and valuable. But in the current submission, this analysis is relatively compressed, and the primary contribution is framed around CSFT as a training method. In its present form, I still find the paper novelty limited.

---

> ### Author Response · Authors · 2025-11-26
>
> The reviewer argues that Lin et al. did not use CoT due to the model landscape in 2022 and further suggests that their method is conceptually applicable to modern reasoning models. **This interpretation is not consistent with the facts.** CoT Prompting was already established at the time of Lin et al., as evidenced by Wei et al. (2022) [1], and the absence of CoT in their work reflects the simple reality that their method was developed for non-reasoning, simple arithmetic tasks. **More importantly, the assertion that Lin et al.’s approach could be applied to contemporary reasoning models remains purely theoretical and is not supported by any empirical evidence.**
>
> In general response (**Table R.1**), we directly implemented Lin et al.’s method under the exact same backbone, dataset, and reasoning conditions as CSFT. Under these matched settings, **Lin et al. did not improve accuracy, did not improve calibration, did not maintain robustness under out-of-distribution evaluation, did not alter their reasoning behavior, and did not exhibit any form of emergent behavior.** By contrast, CSFT consistently improved accuracy and calibration, demonstrated stable out-of-distribution generalization, and produced previously unreported emergent behaviors such as self-verification and numeric-token smoothing. **Treating these two approaches as belonging to the same conceptual category solely on the basis of hypothetical applicability, despite their experimentally contradictory behavior, is not a scientifically grounded comparison.**
>
> This reasoning conflates **theoretical applicability** with **empirical viability.** Scientific novelty is evaluated on what is demonstrably achieved, not on speculative counterfactuals suggesting that a prior method might have achieved the same effect had it been extended. Were such logic applied broadly, many fundamental advances in machine learning—including widely used optimizers and architectural innovations would retroactively be considered non-novel merely because earlier methods could have been modified to approximate them. This does not align with accepted scientific standards.
>
> The reviewer further acknowledges that **instance-level supervision yields substantially stronger calibration and transfer than subtask-level supervision**, and also recognizes that the emergent behaviors we identify are interesting and potentially novel. Yet the review simultaneously concludes that the overall novelty of the work is limited. **These positions cannot both be true.** When a change in supervision granularity fundamentally alters a model’s reasoning process, gives rise to entirely new behaviors, improves both accuracy and calibration, and achieves out-of-distribution generalization that no prior method has demonstrated, the result constitutes a qualitatively new training paradigm rather than an incremental refinement. No existing work, including Lin et al., has shown this combination of effects using simple supervised fine-tuning without any CoT supervision.
>
> ---
>
> [1] Wei, Jason, et al. "Chain-of-thought prompting elicits reasoning in large language models." Advances in neural information processing systems 35 (2022): 24824-24837.

---

> > ### Comment · Reviewer_YHJH · 2025-11-27
> >
> > Thank you for the detailed follow-up. I want to clarify that my concern is not about whether instance-level labels work better than subtask-level labels (your experiments convincingly show that they do) but about how the paper frames its contribution. The manuscript repeatedly presents CSFT as a fundamentally new training method, when in fact the underlying procedure is the same paradigm introduced by Lin et al.  Moving from subtask-level to instance-level labels is an important refinement, but it is still a refinement within that established framework, not a new method and as such, not enough contribution by itself. I believe that the paper needs to be revised to reflect that, beyond the two lines mentioning that previous papers tried something similar, which were added during rebuttal.
> >
> > My assessment is therefore not about the quality or usefulness of your results but about the accuracy of the paper’s claims and the way it presents its findings. In its current form, the framing overstates the novelty of CSFT relative to prior work, and adjusting that framing would make the contribution both clearer and more scientifically grounded.

---

> ### Author Response · Authors · 2025-11-26
>
> **We appreciate the reviewer’s positive assessment that the emergent-behavior analysis could stand as a compelling contribution on its own.** However, the claim that this analysis is compressed does not reflect the breadth of experiments we have already conducted. The paper examines the conditions under which self-verification arises (**Figure 2-b, Figure 3**), compares prefix and suffix supervision (**Figure 4-a**), demonstrates the suppressive effect of KL regularization (**Figure 4-a**), analyzes how the design of confidence labels affects behavior (**Table 7, Table R.1**), evaluates confidence–accuracy relationships through reliability diagrams (Figure 4-b), and investigates how confidence guides error correction during reasoning (**Figure 3**). **This scope is consistent with the structure of prior empirical studies of emergent misalignment [2]**, where the phenomenon was first documented rigorously and only later followed by causal explanatory work [3]. In the same way, our submission provides a reproducible empirical foundation that identifies when, and under which forms of supervision, these behaviors arise, which is the necessary first step before pursuing deeper causal analysis.
>
> If the reviewer is requesting additional analysis of self-verification specifically within our SFT framework, it would be extremely helpful to clarify what form this analysis should take. The current request is not actionable because “analyze why it happens” does not define a concrete experimental objective. We are prepared to conduct further investigation once the reviewer specifies the particular aspect of the behavior that should be examined.
>
> In summary, both the CSFT method and the emergent behaviors it reveals represent clear and substantive contributions. The assessment that the paper has limited novelty is inconsistent with the empirical evidence provided. CSFT is the first method to show that instance-level continuous confidence supervision, without any chain-of-thought supervision, reinforcement learning, or sampling, can fundamentally reshape reasoning behavior, induce new emergent capabilities, and simultaneously improve accuracy and calibration across complex reasoning tasks. No prior work has demonstrated this combination of results.
>
> ---
>
> [2] Betley, Jan, et al. "Emergent Misalignment: Narrow finetuning can produce broadly misaligned LLMs." Forty-second International Conference on Machine Learning.
>
> [3] Wang, Miles, et al. "Persona features control emergent misalignment." arXiv preprint arXiv:2506.19823 (2025).

---

> ### Author Response · Authors · 2025-11-28
>
> We would like to clarify once more that the novelty of our work does not arise from any single design choice in isolation, but from the particular combination of three elements taken together: (1) CSFT operates in complex CoT reasoning settings, (2) it relies solely on simple SFT without any RL or additional sampling, and (3) it is, to our knowledge, the first to empirically show that such minimal supervision without any CoT supervision can simultaneously **improve accuracy and calibration while also triggering new emergent behaviors such as self-verification, CoT-length adjustment, numeric-token smoothing, and transfer to non-reasoning tasks.** This is precisely why our paper is titled “Verbalized Confidence Triggers Emergent Behaviors.”
>
> And we believe the reviewer’s latest comment contains several internal inconsistencies that merit clarification.
>
> The reviewer asserts that CSFT is merely an incremental refinement of Lin et al., yet **our controlled reimplementation (Table R.1) shows that the two methods exhibit fundamentally different empirical behaviors across calibration, accuracy, OOD generalization, reasoning dynamics, and emergent phenomena.** Methods that truly belong to the same training paradigm do not produce such sharply divergent outcomes under identical conditions.
>
> **The reviewer also acknowledges that our emergent behaviors are “interesting enough to form a compelling contribution on their own,” while simultaneously characterizing the underlying training approach as non-novel.** In contemporary machine learning, however, conceptual novelty is grounded not in surface-level procedural differences but in demonstrably distinct model behaviors. Historically, changes far smaller than ours, such as CoT prompting (a prompt phrasing change), LoRA (low-rank updates added to standard fine-tuning), or DPO (a simplified reformulation of RLHF), have been recognized as new methods precisely because they yielded qualitatively different generalization patterns. CSFT is similar in this regard: instance-level continuous labels with suffix positioning and single-pass SFT lead to improvements and behavioral shifts that Lin et al.’s procedure does not reproduce.
>
> Finally, the suggestion that Lin et al. “could in principle” be applied to modern reasoning models is speculative and unsupported; scientific evaluation is based on what has been empirically demonstrated, not on hypothetical extensions of earlier work.
>
> For these reasons, the claim that CSFT is merely incremental does not align with the evidence presented, established ML standards for assessing novelty, or the reviewer’s own acknowledgment of the new behaviors observed. If there are specific aspects of our framing that the reviewer believes should be adjusted, we would greatly appreciate concrete guidance. **At present, the feedback states that the contribution is limited without identifying which precise claims or sections require revision, which makes it difficult to understand what further changes would satisfactorily address the concern.**

---

### Author Response · Authors · 2025-11-19
**General Response**

We thank all reviewers for their valuable and constructive feedback. They noted that the paper is clearly written and easy to follow (**R-YHJH**, **R-7KJt**) and well-motivated in addressing the important challenge of calibration for reasoning models (**R-V8Nd**). Reviewers also highlighted the conceptual insight of using verbalized confidence for self-checking (**R-6NDw**, **R-V8Nd**). They further recognized the novelty of our analysis, including cross-domain generalization, and found the overall presentation informative (**R-YHJH**, **R-7KJt**). Empirically, they found the approach demonstrates strong and generalizable improvements across diverse reasoning and non-reasoning tasks (**R-6NDw**, **R-7KJt**, **R-V8Nd**).

We also thank reviewers **R-YHJH** and **R-V8Nd** for their insightful questions regarding the connection to prior work, particularly Lin et al. (2022). **We believe the concerns partly stem from a misunderstanding of the primary contribution and scope of our CSFT method relative to this prior work.** While Lin et al. (2022) is indeed related through its use of the term verbalized confidence, our CSFT approach differs fundamentally in methodology (instance-level supervision vs. subtask-level labels), scope (general reasoning tasks vs. arithmetic confidence mapping), and outcomes (emergent behaviors).

Most importantly, to our knowledge, our work is the first to show that simple confidence-conditioned supervised fine-tuning, without any reinforcement learning, can induce emergent behaviors such as self-verification in contemporary reasoning-capable LLMs, a distinction not present in prior work.

To clarify these differences, we include (i) a detailed point-by-point comparison in the general response, and (ii) new experimental results (Table R.1) in which we directly implemented Lin et al. (2022) on our benchmarks. We have also incorporated the corresponding analyses and clarifications into a revised manuscript. Please find the details of these updates below.

---

> ### Author Response · Authors · 2025-11-19
> **Clarification of Key Methodological Differences and Addressing Misunderstandings**
>
> # 1. Key Methodological Difference: Instance-Level vs. Task-Level Calibration
>
> ## (a) Lin et al. use subtask-level fixed accuracy labels (i.e., “Task Difficulty”)
>
> Lin et al. assigned a single fixed confidence score to all questions within a subtask $T$ (e.g., "2-digit addition").
>
> $c(q) = \hat{p}_T \quad \text{for all } q \in T,$
>
> This approach relies on two strong assumptions that fail in modern reasoning contexts:
>
> - Homogeneity: It assumes all problems in a subtask share the same difficulty. However, in benchmarks like GSM8K or MATH, a "Geometry" problem can range from a trivial definition check to a complex proof.
> - Greedy-Decoding Bias: It is based on greedy decoding accuracy, which reflects dataset bias rather than the model's internal uncertainty.
>
> **Why does this fail?**: As Lin et al. admitted, their model is forced to output low confidence even for trivial questions like "$10 \times 10 = 100$" simply because the "Multiplication" subtask is generally hard for the model. This is not calibration; it is rather close to category memorization.
>
> ## (b) CSFT uses instance-level stochastic uncertainty (i.e., “Instance-Level Posterior”)
>
> CSFT estimates the true posterior probability of correctness for each *specific input* via consistency sampling ($K$-sampling):
>
> $\hat{p}(q) = \frac{1}{K} \sum_{i=1}^{K} \mathbf{1}[a_0^{(i)} = a^*], \qquad c(q) = 10 \cdot \lfloor 10 \cdot \hat{p}(q) \rfloor .$
>
> **Why is this superior?**: This method captures the model's actual epistemic uncertainty. Even within the same task type, CSFT assigns high confidence to questions the model knows consistently and low confidence to those it finds ambiguous. This teaches the model "Do I know this problem?" rather than "Is this problem type historically hard for me?"
>
> ---
>
> # 2. Beyond Calibration: Confidence-Driven Emergent Reasoning
> The central contribution of our work is not just producing a confidence score, but discovering that **calibrated supervision fundamentally alters the reasoning process.**
>
> Unlike Lin et al., which operated on GPT-3 without leveraging CoT reasoning and focused on simple arithmetic tasks, CSFT induces structural changes in the model's latent representations and generation policy:
>
> - **Confidence-Triggered Self-Correction**: Low confidence causally triggers longer reasoning chains and self-verification phrases (e.g., "Let me double-check..."), as shown in Figure. 3.
> - **Local Token Smoothing**: We observe a geometric smoothing in the token space around confidence values, indicating the model has internalized confidence as a continuous manifold (Figure 2-a).
> - **OOD Generalization**: Because the model learns an internalized epistemic signal (rather than memorizing surface-level task difficulty), it exhibits robust cross-paradigm generalization—successfully transferring from mathematical reasoning (training domain) to non-mathematical and even non-reasoning benchmarks (e.g., Hellaswag, ARC-Challenge, MMLU), whereas Lin et al.’s method fails to transfer (see Table R.1 below).
>
> Lin et al.'s approach, by design, cannot produce these emergent reasoning behaviors as it lacks instance-specific signals and operates without CoT supervision.
>
> ---
>
> # 3. Empirical Comparison: CSFT Exhibits Better Generalization and Calibration
>
> To empirically demonstrate the superiority of our instance-level approach, we re-implemented Lin et al. using their exact protocol (grouping by subtask types in the MATH dataset and assigning average accuracy labels) on Llama-3.2-3B-Instruct.
>
> Lin et al.'s method collapses on OOD tasks (AUROC $\approx$ 50, equivalent to random guessing), confirming that relying on category-level difficulty is insufficient for robust calibration. In contrast, CSFT maintains strong calibration and accuracy across diverse domains, proving that instance-level uncertainty is the correct supervision signal for generalizable calibration.
>
> ---
>
> ### **Table R.1. Comparison between CSFT and Lin et al. (2022) trained on MATH.**
> | Dataset | Method | AUROC | ACC | ECE | Brier |
> |---------|--------|--------|--------|--------|--------|
> | MATH (ID) | CSFT | 78.74 | 30.52 | 0.0324 | 0.1638 |
> | | Lin et al. | 53.82 | 28.18 | 0.1813 | 0.2764 |
> | GSM8K | CSFT | 74.80 | 73.69 | 0.2243 | 0.1805 |
> | | Lin et al. | 50.00 | 67.85 | 0.3215 | 0.3214 |
> | HellaSwag | CSFT | 79.29 | 69.55 | 0.1787 | 0.2564 |
> | | Lin et al. | 47.85 | 49.36 | 0.2572 | 0.3883 |
> | MMLU | CSFT | 56.70 | 61.18 | 0.1420 | 0.2787 |
> | | Lin et al. | 40.73 | 48.44 | 0.3043 | 0.4090 |

---

> ### Author Response · Authors · 2025-11-19
>
> To address the concerns raised regarding the distinction between our work and Lin et al. (2022), and to further clarify our contribution, we have incorporated the following revisions into the updated manuscript:
>
> - We have explicitly discussed Lin et al. and its limitations in the Related Work section (Revised version, Lines 102–105).
>
> - We have added a detailed experimental comparison and methodological clarification in Appendix B.1 to demonstrate the superiority of our instance-level approach empirically.
>
> We sincerely hope that these revisions and the new experimental evidence fully resolve any concerns regarding the novelty and effectiveness of our method. If the reviewers find that our response and revision have adequately addressed the issues, **we would respectfully request that you reconsider your assessment and score.**

---

### Note · Program_Chairs · 2026-01-17
**Submission Desk Rejected by Program Chairs**

The following references in this submission do not refer to real documents and/or have major errors in bibliographic information:

 Bailin Chen, Hanjun Dai Zheng, Zhangir Yang, Alexandre Passos, and Andrew Dai. Program-of-
thoughts prompting: Disentangling computation from reasoning for numerical reasoning tasks.
In Findings of the Association for Computational Linguistics: ACL 2023, pp. 9757–9778, 2023.